# Human cytomegalovirus interactome analysis identifies degradation hubs, domain associations and viral protein functions

Luis V Nobre[1], Katie Nightingale[1], Benjamin J Ravenhill[1], Robin Antrobus[1], Lior Soday[1], Jenna Nichols[2], James A Davies[3], Sepehr Seirafian[3], Eddie CY Wang[3], Andrew J Davison[2], Gavin WG Wilkinson[3], Richard J Stanton[3], Edward L Huttlin[4], Michael P Weekes[1]*

[1]Cambridge Institute for Medical Research, University of Cambridge, Cambridge, United Kingdom; [2]MRC-University of Glasgow Centre for Virus Research, Glasgow, United Kingdom; [3]Division of Infection and Immunity, Cardiff University School of Medicine, Cardiff, United Kingdom; [4]Department of Cell Biology, Harvard Medical School, Boston, United States

**Abstract** Human cytomegalovirus (HCMV) extensively modulates host cells, downregulating >900 human proteins during viral replication and degrading ≥133 proteins shortly after infection. The mechanism of degradation of most host proteins remains unresolved, and the functions of many viral proteins are incompletely characterised. We performed a mass spectrometry-based interactome analysis of 169 tagged, stably-expressed canonical strain Merlin HCMV proteins, and two non-canonical HCMV proteins, in infected cells. This identified a network of >3400 virus-host and >150 virus-virus protein interactions, providing insights into functions for multiple viral genes. Domain analysis predicted binding of the viral UL25 protein to SH3 domains of NCK Adaptor Protein-1. Viral interacting proteins were identified for 31/133 degraded host targets. Finally, the uncharacterised, non-canonical ORFL147C protein was found to interact with elements of the mRNA splicing machinery, and a mutational study suggested its importance in viral replication. The interactome data will be important for future studies of herpesvirus infection.

*For correspondence:
mpw1001@cam.ac.uk

## Introduction

Human cytomegalovirus (HCMV) persistently infects the majority of the worldwide population (*Mocarski et al., 2013*). Following primary infection under the control of a healthy immune system, a latent infection is established that persists lifelong (*Reeves et al., 2005*). In immunocompromised individuals, particularly transplant recipients and AIDS patients, virus reactivated from latency to induce lytic infection is capable of affecting almost any organ system and causing serious disease (*Nichols et al., 2002*). HCMV infection in utero is a leading cause of deafness and intellectual disability in newborns, affecting ~1/200 pregnancies (*Mocarski et al., 2013*).

Small-molecule disruption of critical virus-virus or virus-host protein interactions could provide novel therapeutic strategies. Indeed, disruption of interactions between antiviral restriction factors (ARFs) and viral antagonists can facilitate endogenous inhibition of infection (*Nathans et al., 2008*). Systematic characterisation of all viral protein interactions thus has important implications for antiviral therapy, and is particularly important for HCMV, for which only a few drugs are available.

HCMV encodes 170 canonical protein-coding genes (*Gatherer et al., 2011*), and a substantial number of non-canonical open reading frames (ORFs) that potentially encode additional proteins

have been identified by ribosomal footprinting and proteomics (*Nightingale et al., 2018*; *Stern-Ginossar et al., 2012*). During productive infection in vitro, HCMV gene expression is conventionally divided into immediate-early, early and late phases over a replication cycle lasting ~96 hr. Five temporal classes of viral protein expression have been defined by measuring viral protein profiles over time (*Weekes et al., 2014*). Latent infection with HCMV occurs in a restricted range of cell types, and may involve a somewhat more limited range of viral gene expression (*Goodrum and McWeeney, 2018*; *Schwartz and Stern-Ginossar, 2019*). However, at least some viral proteins function similarly during both productive infection and latency. For example, UL138, which plays roles in the establishment and maintenance of latent infection, downregulates Multidrug Resistance-Associated Protein 1 (MRP1) during both phases of infection (*Weekes et al., 2013*; *Weekes et al., 2014*).

The functions of many canonical HCMV proteins remain poorly understood, and it is not yet clear how many, if any, non-canonical ORFs encode functional polypeptides. We have shown previously that >900 host proteins are downregulated >3 fold over the course of HCMV infection, with 133 proteins degraded in the proteasome or lysosome during the early phase (*Nightingale et al., 2018*; *Weekes et al., 2014*). However, it is not yet known which viral factors target these proteins, and certain proteins, including MHC class I molecules and natural killer cell ligands, can be targeted by more than one viral factor (*Fielding et al., 2014*; *Hsu et al., 2015*; *van der Wal et al., 2002*; *Wilkinson et al., 2008*).

Here, an examination of each canonical and a subset of non-canonical HCMV proteins in infected cells revealed an extensive network of >3400 high confidence virus-host and >150 virus-virus interactions. This provided insights into the functions of multiple uncharacterised or partly characterised viral proteins. The data enabled identification of individual viral factors that target 31 host proteins for degradation. Novel interactions between selected viral and host protein domains were also tested experimentally. In addition, the study provided the first evidence for a functional role for a non-canonical HCMV ORF in viral infection. The extensive interactome data generated in this study predicts viral proteins important in key cellular pathways, and may lead to the development of new antiviral therapeutics.

## Results

### Construction of the HCMV-host interactome

To build a global picture of all HCMV virus-host and virus-virus protein interactions, 170 stable cell lines were generated from immortalised primary human fetal foreskin fibroblasts (HFFF-TERTs), each expressing a single, canonical HCMV ORF with a C-terminal V5 tag to facilitate immunoprecipitation (IP). Two non-canonical ORFs, ORFL147C and ORFS343C, were also included on the basis of either high or low expression respectively, relative to all other viral ORFs detected previously by proteomics (*Figure 1 – Figure Supplement 1A*, *Supplementary file 1A*) (*Fielding et al., 2017*; *Weekes et al., 2014*). Prior to profiling by IP-mass spectrometry (IP-MS), expression of each tagged viral 'bait' protein was validated by immunoblotting (IB), MS or RT-qPCR, apart from UL136 which could not be detected by any method (*Figure 1 – Figure Supplement 1B*, *Supplementary file 1B*). To examine the full range of virus-virus interactions in addition to virus-host interactions, IP was performed in cells infected with Merlin strain HCMV at multiplicity of infection (MOI) of 2 for 60 hr. Merlin contains a full length genome and expresses all HCMV genes apart from UL128 and RL13. All detectable viral proteins are expressed at 60 hr post-infection (PI) with this strain (*Weekes et al., 2014*) (*Figure 1—figure supplement 1E*). A schematic and details of the IP-MS strategy are shown in *Figure 1*.

For HCMV UL120 and UL142, no interacting proteins passed the stringent filters employed. For seven further proteins, only the bait itself passed filtering, leaving 162 viral baits with ≥1 HCIP. In total, 3572 interactions were detected across all 162 baits, with a range of 1–174 interactions per bait, reflecting a scale-free degree distribution typical of protein interaction networks. The median number of interactions per bait was 9, similar to previously observed in the Bioplex 2.0 human interactome (*Huttlin et al., 2015*) (Materials and methods; *Supplementary file 2A*, *Figure 1—figure supplement 2A*). Data were validated from previously reported virus-virus and virus-host interactions described in BioGRID, IntAct, Uniprot, MINT and Virus Mentha (*Figure 1—figure supplement 2B*,

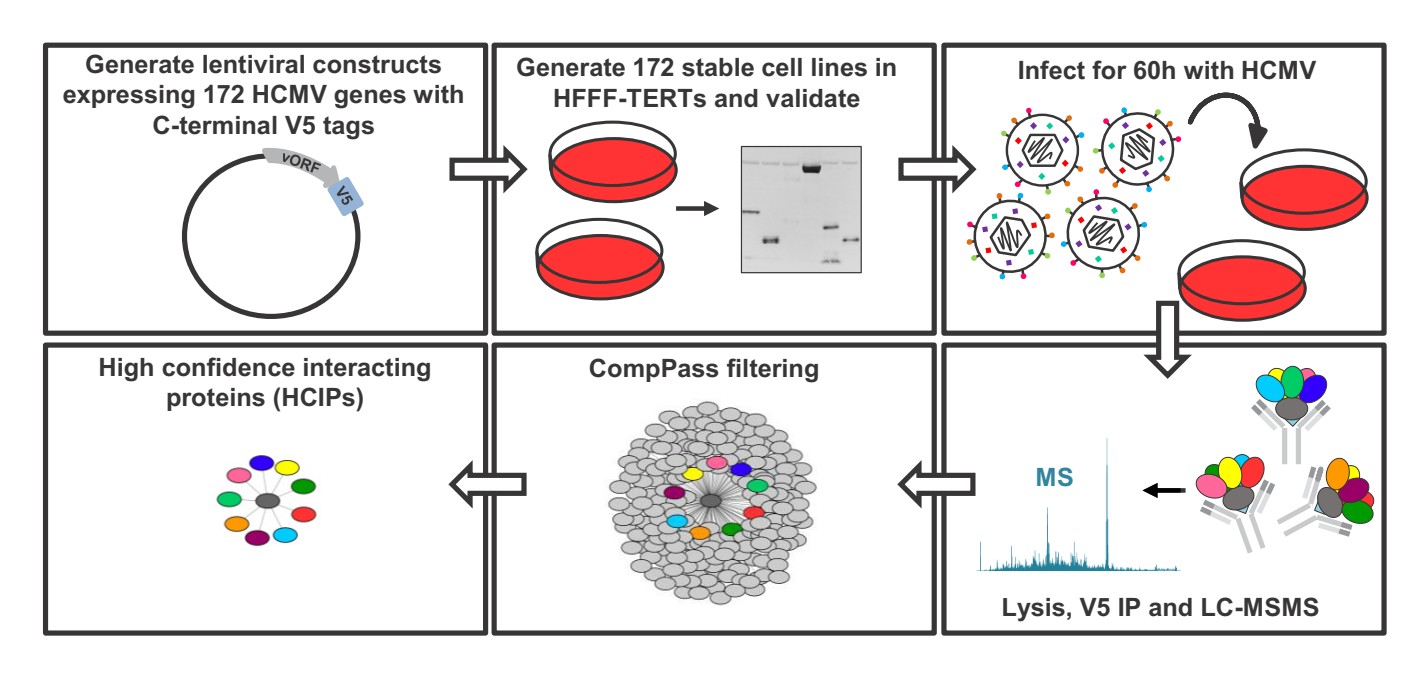

**Figure 1.** Schematic of the IP strategy. IP samples were generated and analysed in technical duplicate, using the method originally described in *Huttlin et al. (2017)*; *Huttlin et al. (2015)* and discussed in detail in the Materials and methods section. For 153 baits with zero or one transmembrane (TM) region predicted by Uniprot, an NP40-based lysis buffer was used; for 18 baits with >1 TM region, a digitonin-based buffer was used, as this has previously been demonstrated to improve identifications of interacting proteins ('prey') (*Babu et al., 2012*) (*Supplementary file 1B*). Each dataset was scored separately using the CompPASS algorithm (*Huttlin et al., 2015*; *Sowa et al., 2009*) to better model detergent-specific variation in IP-MS background. Data reported for each prey protein in every IP include: (a) the number of peptide spectral matches (PSMs), averaged between technical replicates; (b) an entropy score, which compares the number of PSMs between replicates to eliminate proteins that are not detected consistently; (c) a z-score, calculated in comparison to the average and standard deviation of PSMs observed across all IPs; and (d) a normalised WD (NWD) score. The NWD score addresses whether (i) the protein is detected across all IPs, and (ii) whether it is detected reproducibly among replicates. It was calculated as described in *Behrends et al. (2010)* using the fraction of runs in which a protein was observed, the observed number of PSMs, the average and standard deviation of PSMs observed for that protein across all IPs, and the number of replicates (1 or 2) containing the protein of interest. NWD scores were normalised so that the top 2% earned scores of ≥1.0. Stringent filters were applied to remove inconsistent and low-confidence protein identifications across all IPs and thus minimise both false protein identifications and associations (*Huttlin et al., 2015*). These included: (a) a minimum PSM score of 1.5 (i.e. ≥3 peptides per protein across both replicates); (b) an entropy score of ≥0.75; and (c) an NWD or z-score in the top 2%. Previous studies have estimated a 5% false discovery rate when employing a similar strategy with a top NWD score cutoff of 2% (*Sowa et al., 2009*). Interactions passing these criteria are named 'high confidence interacting proteins' (HCIPs) (*Supplementary file 2B*), and were used in all subsequent analyses. For added stringency, the supervised learning algorithm CompPass Plus was employed. This additionally assessed batch variations, overall spectral counts, unique peptide counts and protein detection frequency. Shannon entropy quantified a protein's consistency of detection across technical duplicate LC-MS analyses, removing inconsistent protein identifications (*Huttlin et al., 2017*). CompPass plus was developed for interactomes with ≥96 baits and in the present study was only applied to the 153 baits solubilized in NP40. Interactions that passed CompPass filters, had CompPass Plus p(Interactor) >0.75 and in which the prey was identified by ≥2 unique peptides were considered as very high confidence interacting proteins (VHCIPs). These are indicated in green shading in *Supplementary file 2B*. To facilitate global analysis of all data, and because digitonin-solubilised interactions were not analysed using CompPass plus, HCIPs as opposed to VHCIPs were examined for the remainder of this study. The identification of an interacting protein as a VHCIP nevertheless adds additional confidence that the interaction observed is likely to be genuine.

The online version of this article includes the following source data and figure supplement(s) for figure 1:

**Figure supplement 1.** Further details of the interactome.
**Figure supplement 1—source data 1.** Correlation of the number of total, unique and bait peptides from each protein identified in replicates 1 and 2.
**Figure supplement 1—source data 2.** Reproducibility of interactome analyses.
**Figure supplement 2.** Further details of interactions.

*Supplementary files 2–3*) (*Calderone et al., 2015*; *Chatr-Aryamontri et al., 2013*; *Licata et al., 2012*; *Orchard et al., 2014*).

## Systematic analysis of viral protein function

Systematic analysis of protein interactions can improve understanding of viral protein function. To analyse the functions of all viral proteins simultaneously, DAVID software (*Huang et al., 2009*) was employed to determine which pathways were enriched amongst the 3416 human proteins that interacted with viral baits (*Figure 2* centre, *Figure 2—figure supplement 1*, *Supplementary file 4A-B*).

Nucleosome remodeling (NuRD) complex components were significantly enriched among HCMV-interacting proteins. The NuRD complex plays major roles in cellular chromatin remodeling, and is known to be co-opted by HCMV UL29 and UL38 to enhance expression of immediate-early genes (*Savaryn et al., 2013*; *Terhune et al., 2010*). The interaction of UL29 and UL38 in a complex with all components of NuRD was confirmed, in addition to p53 (*Savaryn et al., 2013*). UL29 was also found to interact with multiple human proteins that function in histone deacetylation, which had not been observed previously (*Figure 2*).

UL87, UL79, UL91 and UL95 are essential for viral replication and necessary for transcriptional activation of viral genes expressed with 'true late' kinetics. UL92 has a similar function, and it has been suggested that these five proteins may form one or more complexes that modulate RNA polymerase II activity (*Isomura et al., 2011*; *Omoto and Mocarski, 2013*; *Omoto and Mocarski, 2014*). Interactome data confirmed that UL87 interacted with UL79, UL91 and UL95 but did not detect an interaction with UL92. This latter observation, and in fact the lack of identification of any viral-viral UL92 interactions, may be explained by our finding that UL92 was one of the two least abundantly expressed viral proteins during HCMV infection (*Supplementary file 1A*, bottom). UL87 also interacted with all 12 components of the RNA polymerase II (RPII) complex and the associated protein RPII Associated Protein 2 (RPAP2) (*Figure 2*). The UL87-RPII interaction was anticipated by analogy to the orthologous RPII-interacting Epstein-Barr virus protein BcRF1, but had not previously been demonstrated. Interaction of UL87, UL95 and UL79 with the UL97 protein kinase was also novel.

Collectively, these confirmatory data indicate that the HCMV interactome has the power to predict new functions for uncharacterised or partly characterised viral proteins, particularly where a bait interacts with multiple protein components of the same pathway. For example, UL72 is a temporal protein profile 3 (Tp3)-class HCMV protein derived from deoxyuridine 5'-triphosphate nucleotidohydrolase (dUTPase) in other herpesviruses, but lacks dUTPase activity (*Caposio et al., 2004*; *McGeehan et al., 2001*). UL72 interacted with all 10 components of the CCR4-NOT (carbon catabolite repressor 4-negative on TATA) complex, which is a key regulator of gene expression from production of mRNAs in the nucleus to their degradation in the cytoplasm (*Yi et al., 2018*). The interaction between UL72 and CNOT2/CNOT7 was confirmed by co-IP (*Figure 3A–B*). It remains to be determined how UL72 modulates CCR4-NOT function.

The hitherto uncharacterised viral UL145 protein is known to recruit the Cullin 4 E3 ligase scaffold and associated adaptor proteins, and to degrade helicase-like transcription factor (HLTF) (*Nightingale et al., 2018*). Interactome data suggested that all human proteins interacting with UL145 and the paralogous RL1 were part of the ubiquitin conjugation pathway (*Supplementary file 2*, *Supplementary file 4*), and furthermore that RL1 interacted with Cullin 4 (CUL4, *Figure 2*). The interaction with CUL4A was validated by co-IP (*Figure 3C*). Proteins that are degraded after binding RL1/CUL4 still require identification; it is possible that their abundance after degradation may have been insufficient to enable identification in this study. Multiple other HCMV proteins additionally interacted with elements of the ubiquitin transfer or conjugation pathways, including the inhibitor of apoptosis UL36, which bound the Cullin one scaffold, E3 ligase UBR5, and F-box component FBOX3. Similarly, DNA helicase/primase component UL102 interacted with E3 ligase RNF114 and E2 conjugating enzyme UBE2L6 (*Figure 2* and *Supplementary file 2*).

The tegument protein UL71 has an essential function in the final steps of secondary envelopment leading to infectious viral particles, but is expressed with Tp3 kinetics, suggesting the possibility of a role earlier during infection (*Dietz et al., 2018*; *Meissner et al., 2012*; *Weekes et al., 2014*). UL71 interacted with multiple interferon-stimulated proteins (*Figure 3D*), including TRIM22, which restricts replication of HIV-1, influenza A and hepatitis B and C viruses (*Lian and Sun, 2017*). The UL71-TRIM22 interaction was validated by co-IP, suggesting that investigation of a putative innate immune role for UL71 will be important (*Figure 3E*).

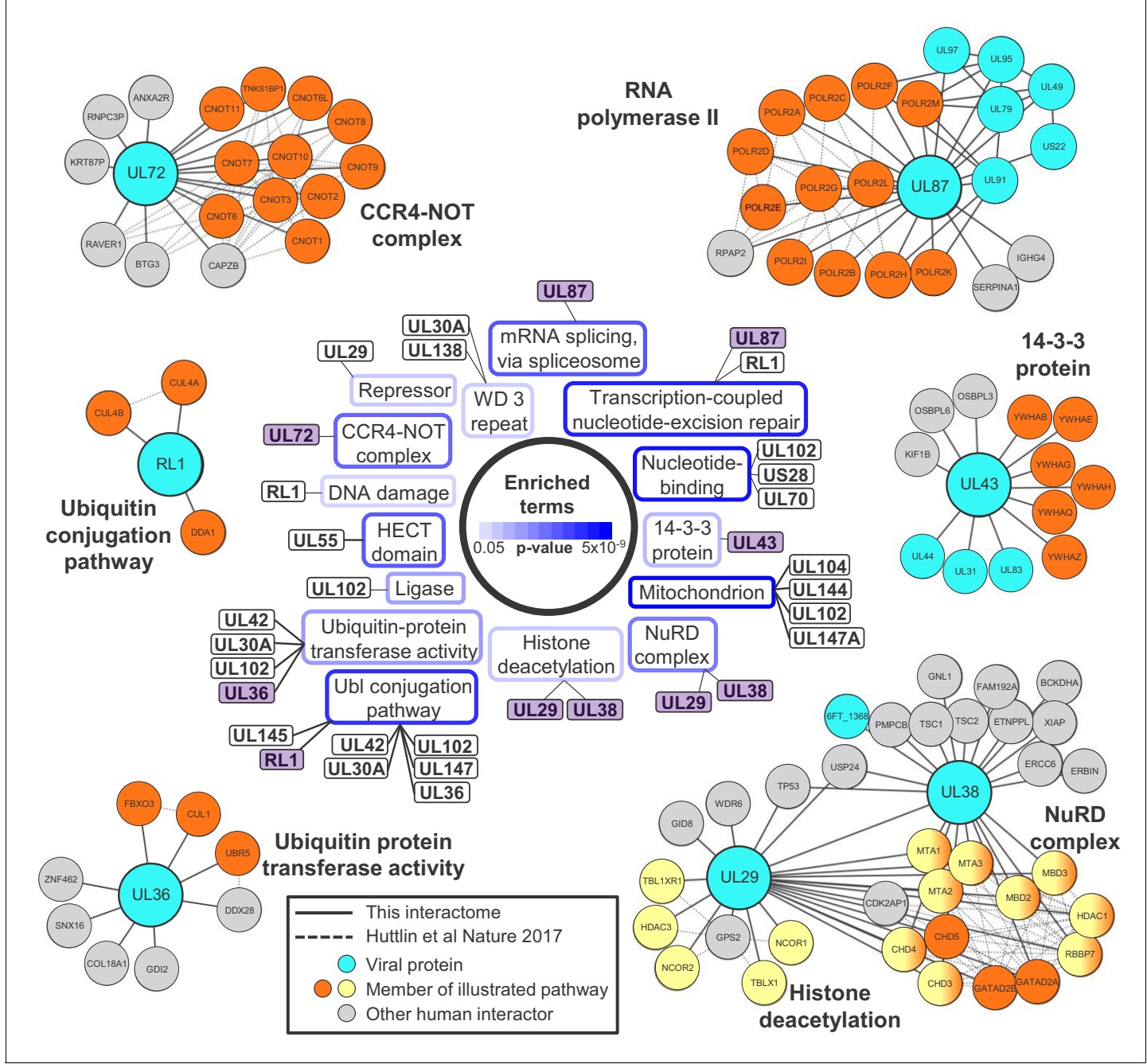

**Figure 2.** Systematic analysis of interactome data predicts novel functions for viral proteins. DAVID software with default settings (*Huang et al., 2009*) was applied to determine which pathways were enriched amongst all HCIPs in the interactome, in comparison to all human proteins as background. Benjamini-Hochberg adjusted p-values are shown as blue surrounds to each pathway enriched at p<0.05. Viral baits are linked to enriched pathways where > 33% of human interacting proteins belonged to a given pathway, and examples are shown around the outside of the figure. These examples are indicated in the central part of the figure by purple shading. For example, 6/9 (67%) human HCIPs for UL43 were part of the 14-3-3 protein family. Viral baits are shown as large turquoise circles, and interacting viral proteins as smaller turquoise circles. Members of enriched pathways are shown in orange or yellow (for NuRD complex and histone deacetylation, protein membership of both pathways is indicated by half-orange, half-yellow circles). Solid lines indicate interactions identified by this interactome, and dashed lines indicated interactions derived from human Bioplex 2.0 and subsequent unpublished data (*Huttlin et al., 2017* and http://bioplex.hms.harvard.edu/downloadInteractions.php). Full data are shown in *Supplementary file 4*. As an alternative approach to highlight cellular functions that predominantly related to individual viral proteins, *Figure 2—figure supplement 1* shows pathways with p<0.05 (after Benjamini-Hochberg adjustment) and for which > 33% of the identified cellular protein members of the pathway interacted with a given viral bait.

The online version of this article includes the following figure supplement(s) for figure 2:

*Figure 2 continued on next page*

*Figure 2 continued*

**Figure supplement 1.** Pathways enriched with p<0.05 (after Benjamini-Hochberg adjustment) and for which > 33% of the identified components interacted with a given viral bait.

**Figure supplement 2.** Further details of interactions according to viral protein temporal class.

In addition to characterising baits that interacted with multiple members of individual cellular pathways, an alternative approach identified pathways whose members interacted predominantly with single baits (*Figure 2—figure supplement 1*). The US28 G-protein coupled receptor (GPCR) functions in both lytic and latent HCMV infection via constitutive signaling to activate distinct intracellular pathways (*Krishna et al., 2018*). Here, US28 interacted with all quantified members of thick filament/muscle myosin complexes, namely myosin heavy and light chain components, a myosin binding protein and titin. This suggests an unanticipated role for US28 in processes such as regulation of the actin cytoskeleton or cytoskeletal remodeling (*Wang et al., 2018*). Other viral proteins may have novel functions modulating vesicular transport. For example, the US27 GPCR interacted with multiple components of the SNARE complex, whose primary function is to mediate vesicle fusion (*Han et al., 2017*). Envelope glycoprotein UL132 interacted with the AP-2 adaptor complex, which functions in clathrin-mediated endocytosis (*Figure 2—figure supplement 1*) (*Collins et al., 2002*).

To gain further insights into temporal regulation of protein-protein interactions, we determined which functions were enriched amongst human HCIPs for each of the five temporal classes of HCMV bait (*Weekes et al., 2014*). A clear relation to functions required at different stages of the viral lifecycle was observed (*Figure 2—figure supplement 2A*, *Supplementary file 4C*). For example, Tp1 and Tp2 protein HCIPs were enriched in NuRD complex members, proteins involved in histone deacetylation and proteins with SANT domains (which function in chromatin remodelling). Tp3 HCIPs were enriched in functions required for viral genomic replication and immune evasion, whilst Tp5 HCIPs were directed at intracellular trafficking and secretion (*Figure 2—figure supplement 2A*). For viral-viral protein interactions, two patterns emerged – (a) interaction of viral proteins within the same temporal class, or between adjacent classes; (b) interaction of proteins from the largest class (Tp5) with members of each of the five classes (*Figure 2—figure supplement 2B*, *Supplementary file 4D*). For example, Tp1 and Tp2 class proteins UL29 and UL38 interacted, as previously reported (*Supplementary file 3*, *Figure 2*). Tp1-class tegument proteins US23 and US24 interacted. The majority of Tp5 interactions were with other Tp5 proteins, 15/37 of which were tegument-tegument, capsid-capsid or tegument-capsid protein interactions (*Figure 2—figure supplement 2B*). Certain interactions between proteins in different temporal classes have also been reported; for example, between the Tp5 DNA polymerase accessory protein UL44 and Tp2 DNA polymerase UL54. Clearly, other novel interactions also exist between quite distinctly expressed proteins, for example between the functionally unknown Tp2-class membrane protein UL14 and two Tp5-class proteins: membrane protein UL121 and envelope glycoprotein UL4.

## Association between functional domains revealed by protein-protein interactions

Certain domains perform related functions within diverse proteins, often via interactions with complementary structures. The function and interaction(s) of these domains can be predicted by analysing interactions between their parent proteins (*Finn et al., 2014*; *Huttlin et al., 2015*). Although domains that co-occur frequently do not necessarily interact directly, these associations can nevertheless provide insights into domain biology.

By mapping Pfam domains to every bait and prey protein in the interactome, it was possible to identify domain pairs that interact with unusual frequency (*Figure 4A*) (*Finn et al., 2014*). This correctly predicted that HCMV glycoprotein UL141 interacts with TNFR cysteine-rich domains (TNFR c6), which has been demonstrated for TNFRSF10B and predicted for TNFRSF10A (*Nemčovičová et al., 2013*). UL141 also interacted with TNFRSF10D as reported (*Smith et al., 2013*) and was found to interact with TNFRSF1A, suggesting that these interactions may also occur via the TNFR c6 domain (*Figure 4A*, *Supplementary file 5B*).

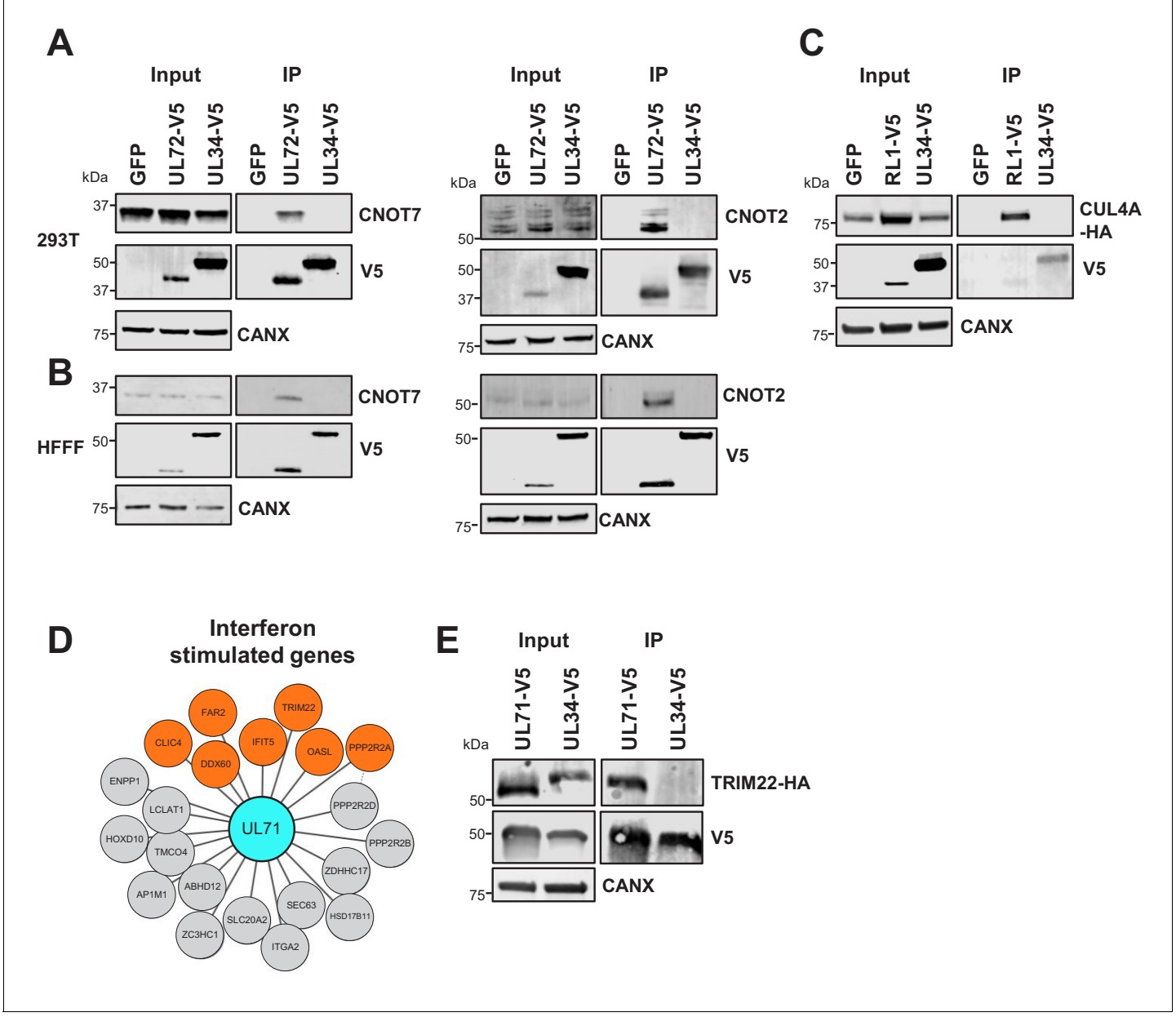

**Figure 3.** Validation of interactome data by co-IP. (**A**) Co-IPs validating that UL72 interacts with CCR4-NOT Transcription Complex Subunits 7 and 2 (CNOT7 and CNOT2), conducted in HEK293T cells. For all experiments in this figure, left panels show an IB of 1–2% of input sample, and right panels shown an anti-V5 co-IP. Cells were transiently transfected with two plasmids, one expressing the C-terminally V5-tagged viral protein and the other expressing the C-terminally HA-tagged cellular prey. Bait proteins were detected with anti-V5, and prey with antibodies against CNOT7 or CNOT2 protein. Controls included GFP or the viral UL34 protein. CANX – calnexin loading control. This figure is representative of n = 1 experiment (CNOT2); n = 2 experiments (CNOT7). Expected sizes: CNOT7: 33 kDa; CNOT2: 52 kDa; CANX: 72 kDa; UL72: 44 kDa; UL34: 45 kDa. (**B**) Co-IPs validating that UL72 interacts with CNOT7 and CNOT2, conducted in HFFF-TERT cells overexpressing C-terminally V5-tagged UL72. Proteins were detected as described in (**A**). This figure is representative of n = 2 experiments (CNOT2); n = 1 experiment (CNOT7). Expected sizes: CNOT7: 33 kDa; CNOT2: 52 kDa; CANX: 72 kDa; UL72: 44 kDa; UL34: 45 kDa. (**C**) Co-IP validating the interaction between RL1 and CUL4A, conducted in HEK293T cells as described in (**A**), but with detection of CUL4A using anti-HA. This figure is representative of n = 4 experiments. Expected sizes: CUL4A: 77 kDa; RL1: 35 kDa; UL34: 45 kDa; CANX: 72 kDa. (**D**) HCMV UL71 interacted with multiple interferon-stimulated proteins, including TRIM22. (**E**) Co-IP validating the interaction between UL71 and TRIM22, conducted as described in (**C**). This figure is representative of n = 3 experiments. Expected sizes: TRIM22: 56 kDa; UL71: 40 kDa; UL34: 45 kDa; CANX: 72 kDa.

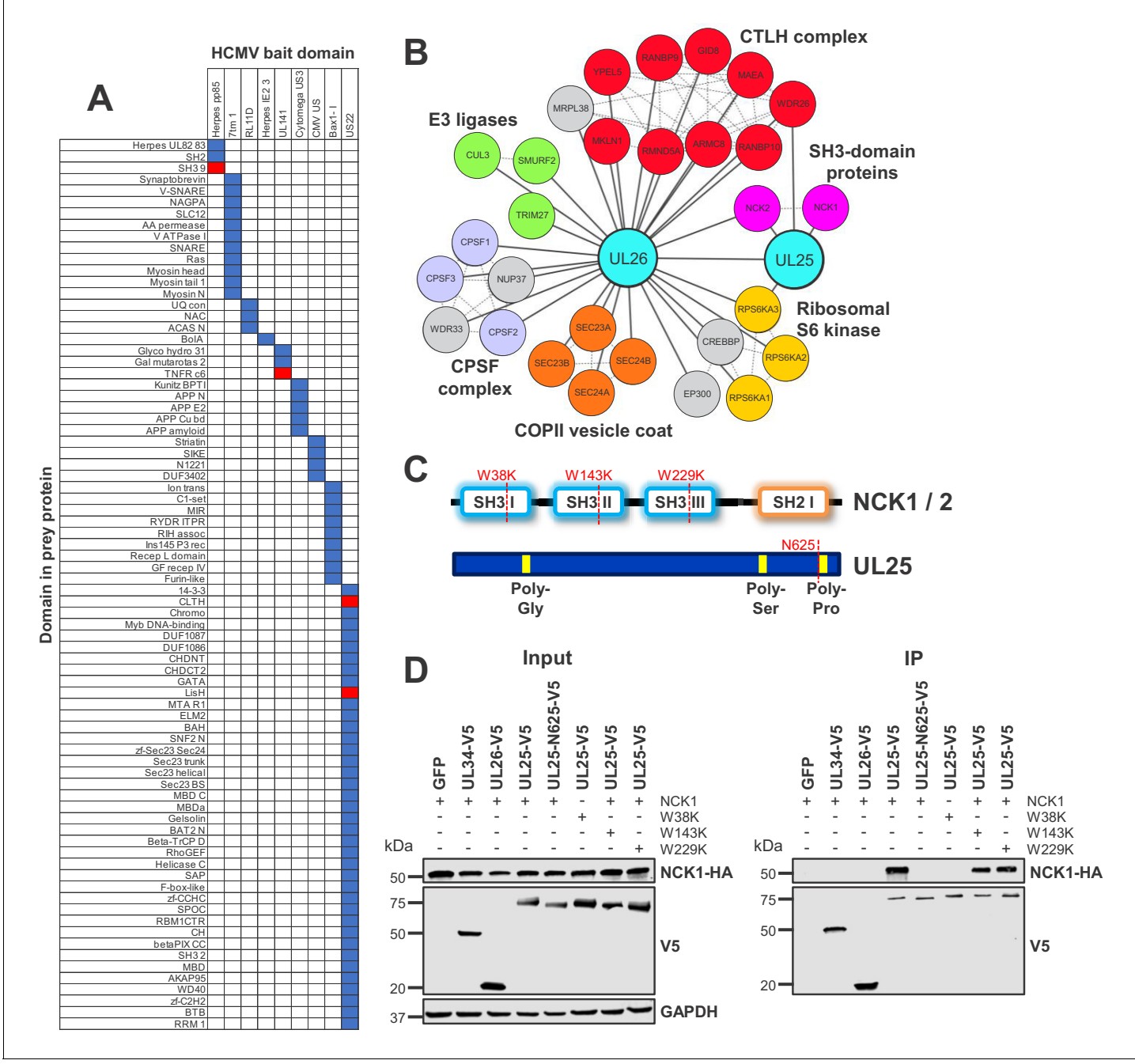

**Figure 4.** Interaction between UL25 and NCK1 identified by domain association analysis. (**A**) Table depicting significant associations between domains present in HCMV baits (top) and human or viral prey (side). Pfam domains were mapped onto every bait and prey protein in the interactome (*Finn et al., 2014*). The numbers of interactions emanating from proteins containing each domain were tallied individually, along with the numbers of interactions linking each observed domain pair. Contingency tables were then populated to relate domain associations. For each pair, Fisher's exact test determined the likelihood of a non-random association. p values were adjusted for multiple hypothesis testing (*Benjamini and Hochberg, 1995*). Coloured boxes identify domain pairs that associate at a 1% false discovery rate (FDR). Red boxes indicate domain pairs from this analysis discussed in the text. Domain associations are only shown for domains occurring in at least two viral proteins. *Supplementary file 5* shows the full underlying data. (**B**) All HCIPs for UL25 and a subset of HCIPs for UL26 (full data are shown in *Figure 4—figure supplement 1*). DAVID analysis identified that members of the C-terminal to LisH (CTLH) complex and COPII vesicle coat proteins were enriched among UL26 HCIPs (*Figure 2—figure supplement 1*). Domain association analysis suggested that interaction of UL26 with CTLH components may occur via interaction of the viral US22 domain with either cellular CLTH or LisH domains (*Supplementary file 5*). Dashed lines represent human-human interactions derived either from Bioplex 2.0 as described in *Figure 2* or from curated or experimental data in the STRING database. CPSF - Cleavage and polyadenylation specificity factor. (**C**) Schematic of NCK1 and UL25 protein structures, indicating the position of point mutations or truncation for (**D**). (**D**) Co-IP demonstrating that the UL25 proline-rich

*Figure 4 continued on next page*

*Figure 4 continued*

C-terminal domain associates with the first NCK1 SH3 domain, conducted as described in *Figure 3*. HEK293T cells were transiently transfected with the indicated plasmids, one expressing the C-terminally V5-tagged viral protein and the other expressing C-terminally HA-tagged NCK1. These proteins were detected with anti-V5 and anti-HA. Mutations or truncations of each gene are indicated in the figure and in (C). GAPDH – loading control. This figure is representative of n = 3 experiments. Expected sizes: NCK1: 43 kDa; UL25: 74 kDa; UL26: 21 kDa; GAPDH: 36 kDa.

The online version of this article includes the following figure supplement(s) for figure 4:

**Figure supplement 1.** Full interaction data for UL25 and UL26, annotated as described in *Figure 4B*.

Domain analysis predicted that certain Herpes pp85 proteins interact with host SH3 domains. Underlying interactome data suggested that the viral tegument pp85 phosphoprotein UL25 interacted with SH3 domain-containing proteins NCK1 (Non-catalytic region of protein tyrosine kinase 1) and NCK2. Additionally, UL25 interacted with two other human proteins and the viral tegument protein UL26. UL26 had more diverse targets, including NCK2 but not NCK1 (*Figure 4A–B*, *Supplementary file 2*, *Supplementary file 5*).

SH3 domains are known to interact with proline-rich regions (*Kurochkina and Guha, 2013*). UL25 has a proline-rich C-terminus, and NCK1 has three N-terminal SH3 regions. A series of mutations or truncations (*Figure 4C*) suggested that the UL25 C-terminus interacts with the first NCK1 SH3 domain alone, validating and extending the prediction from domain association analysis (*Figure 4D*).

NCK1 is a multifunctional cytoplasmic adaptor protein with known roles in signal transduction from receptor tyrosine kinases, cytoplasmic remodeling via regulation of actin polymerization, apoptosis and the DNA damage response (*Buvall et al., 2013*; *Keyvani Chahi et al., 2016*; *Ngoenkam et al., 2014*). Interaction of UL25 with NCK1 may thus fulfill a variety of functions. One possibility may include inhibition of immune synapse formation. HCMV UL135 is known to dispel association between F-actin filaments in target cells and the immune synapse (*Stanton et al., 2014*). UL25 might regulate actin polymerisation in a complementary manner in order to achieve a similar aim.

## Viral proteins that degrade cellular prey

We previously described a multiplexed approach for discovering proteins that have innate immune function on the basis of their active degradation by the proteasome or lysosome during the early phase of HCMV infection. Using three orthogonal proteomic/transcriptomic screens to quantify protein degradation, 133 proteins were shown to be degraded in the proteasome or lysosome during early phase infection, which were enriched in novel antiviral restriction factors (*Nightingale et al., 2018*). To facilitate the mapping of viral gene functions, a final screen employed a panel of HCMV mutants, each deleted in contiguous gene blocks dispensable for virus replication in vitro. However, this screen did not confidently identify the genetic loci that targeted 121/133 degraded proteins. Furthermore, even for 12/133 confidently identified loci, characterization of which individual viral genes degraded cellular targets often proved arduous. For example, to identify UL145 as the gene within the UL133-UL150 block that targeted HLTF to the proteasome, 19 single viral gene deletion mutants required testing (*Nightingale et al., 2018*).

Interactome data revealed viral baits for 31/133 degraded prey (*Supplementary file 6*). The ubiquitin E3 ligase ITCH (Itchy E3 Ubiquitin Protein Ligase) is known to be targeted for degradation by viral UL42 (*Koshizuka et al., 2016*). In addition to ITCH, UL42 interacted with Neural Precursor Cell Expressed, Developmentally Down-Regulated 4 (NEDD4)- family E3 ligases NEDD4 and NEDD4-like (NEDD4L), which were degraded during early HCMV infection (*Figure 5A–B*) (*Nightingale et al., 2018*). These interactions were validated by co-IP using both C- and N-terminally V5 tagged UL42, and UL42 was shown to be sufficient for degradation of NEDD4 (*Figure 5D–E*, *Figure 5—figure supplement 1*). UL42 protein has not been detected in any of our previous proteomic studies (*Fielding et al., 2017*; *Nightingale et al., 2018*; *Weekes et al., 2014*), however UL42 transcript was quantified by Stern-Ginossar et al (*Stern-Ginossar et al., 2012*). Although expression of this transcript peaked at 72 hr of infection, it was nevertheless clearly detectable at early time points, suggesting that UL42 protein is likely to be expressed contemporaneously with degradation of NEDD4 and NEDD4L (*Figure 5C*). The route of degradation of each of the UL42 targets requires further characterisation. MG132 and leupeptin both inhibited degradation of each protein (*Figure 5B*),

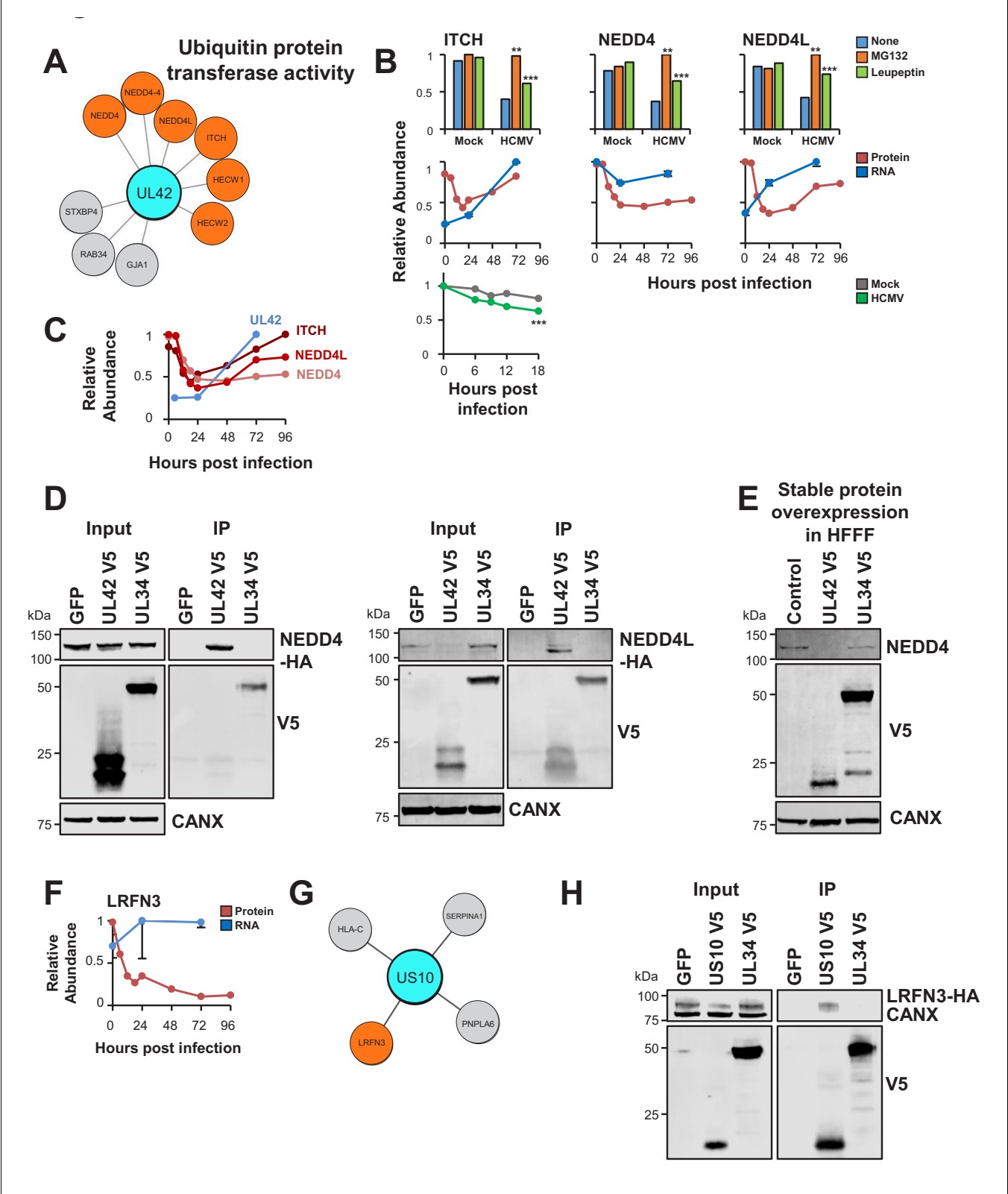

**Figure 5.** UL42 identified as a hub of E3 destruction by a combination of interactome and degradation data. US10 interacts with LRFN3, which is rapidly downregulated from the PM during HCMV infection. (**A**) High-confidence cellular interactors of UL42. 57% of UL42 interactors exhibited ubiquitin protein transferase activity (*Figure 2*, counting NEDD4 only once). UL42 interacted with NEDD4, NEDD4 isoform four and NEDD4L, in addition to HECT, C2 and WW Domain Containing E3 Ubiquitin Protein Ligases HECW1 and 2. NEDD4-4: isoform 4 of NEDD4. (**B**) ITCH, NEDD4 and

*Figure 5 continued on next page*

*Figure 5 continued*

NEDD4L are degraded during early HCMV infection (data from *Nightingale et al., 2018*). Protein degradation was measured using three orthogonal tandem mass tag (TMT)-based proteomic screens. The first measured protein abundance throughout early infection in the presence or absence of inhibitors of the proteasome or lysosome. The second compared transcript and protein abundance over time to distinguish between degraded and transcriptionally regulated proteins. The third employed an unbiased global pulse-chase to compare the rates of protein degradation during HCMV infection against mock infection (NEDD4 and NEDD4L were not quantified in this latter screen). Benjamini-Hochberg adjusted Significance A values were used to estimate p-values in the top panels; **p<0.005, ***p<0.0005. Mean and SEM are shown for transcript quantitation (n = 3) in the middle panels. A p-value for the difference between rates of degradation is shown in the bottom panel; ***p<0.0005. All calculations and statistics are described in *Nightingale et al. (2018)*. (C) UL42 transcript is expressed contemporaneously with NEDD4 and NEDD4L degradation. Protein profiles from *Figure 5B* (red colour, *Nightingale et al., 2018*) are overlaid with a UL42 transcript profile (blue colour, *Stern-Ginossar et al., 2012*). UL42 transcript was not detected in our previous RNAseq analysis (*Nightingale et al., 2018*). (D) Validation of interaction between UL42 and NEDD4 (left panel) and NEDD4L (right panel) by co-IP, conducted as described in *Figure 3*. HEK293T cells were transiently transfected with the indicated plasmids, one expressing the C-terminally V5-tagged viral protein and the other expressing C-terminally HA-tagged NEDD4 or NEDD4L. These proteins were detected with anti-V5 and anti-HA. This figure is representative of n = 2 experiments (NEDD4); n = 1 experiment (NEDD4L). Expected sizes: NEDD4: 104–149 kDa; NEDD4L: 96–111 kDa; UL42: 14 kDa; UL34: 45 kDa; CANX: 72 kDa. (E) UL42 was sufficient to degrade NEDD4. HFFF-TERTs expressing UL42 or controls were lysed and immunoblotted as indicated. Anti-NEDD4 was used to detect endogenous NEDD4. This figure is representative of n = 1 experiment. Expected sizes: NEDD4: 104–149 kDa; UL42: 14 kDa; UL34: 45 kDa; CANX: 72 kDa. (F) LRFN3 was rapidly downregulated from the PM during HCMV infection, in the presence of upregulated transcript (mean and SEM are shown for transcript quantitation (n = 3); data are from *Nightingale et al., 2018*). (G) HCIPs of US10, including LRFN3. (H) Validation of the interaction between US10 and LRFN3 by co-IP, conducted as described in *Figure 3*. Prey were detected using anti-HA. This figure is representative of n = 2 experiments. Expected sizes: LRFN3: 66 kDa; US10: 21 kDa; UL34: 45 kDa; CANX: 72 kDa.

The online version of this article includes the following figure supplement(s) for figure 5:

**Figure supplement 1.** Validation of interaction between UL42 and NEDD4 (left panel) and NEDD4L (right panel) by co-IP, conducted as described in *Figure 3*.

---

which may correspond to the known effects of MG132 on lysosomal cathepsins in addition to the proteasome (*Wiertz et al., 1996*), or effects of leupeptin on certain proteasomal proteases in addition to lysosomal proteases.

To test the sensitivity of the interactome for detecting interactions with weakly-expressed prey, cell surface adhesion molecule Leucine Rich Repeat And Fibronectin Type III Domain Containing 3 (LRFN3) was examined. This protein was previously quantified by a single peptide in samples enriched for plasma membrane (PM) proteins only (*Nightingale et al., 2018*; *Weekes et al., 2014*). LRFN3 was rapidly downregulated from the PM, accompanied by upregulation of transcript over the same period, suggesting either degradation or retention within the infected cell (*Figure 5F*). Only the ER-resident transmembrane glycoprotein US10 interacted with LRFN3, and this was validated by co-IP (*Figure 5G–H*). US10 may downregulate this cell surface molecule in a manner similar to the reported degradation of HLA-G (*Park et al., 2010*).

## ORFL147C is a novel viral protein required for viral replication

It had hitherto been unclear whether any of the 604 HCMV ORFs identified by ribosome profiling (RP-ORFs) encoded functional polypeptides (*Stern-Ginossar et al., 2012*). The abundance of the two RP-ORFs examined in this interactome was in the same range as canonical HCMV proteins, with ORFL147C present at ~25 x lower copy number than the most abundant tegument protein UL83 and ~275 x higher copy number than the membrane protein US18. ORFS343C was ~3 x more abundant than US18 (*Figure 1—figure supplement 1A*). ORFL147C had 80 human HCIPs and ORFS343C 23 human HCIPs (*Supplementary file 2*).

The coding sequence of ORFL147C is oriented parallel to the 5' end of UL56 (*Figure 6A*), which is a canonical gene encoding a subunit of terminase. ORFL147C is expressed with Tp4 kinetics (*Figure 6B*). Enrichment analysis of ORFL147C HCIPs suggested functions in RNA binding, mRNA splicing or transcription (*Figure 6C–D*). We validated the interaction of ORFL147C with Muscleblind Like Splicing Regulator 1 (MBNL1) and CUG Triplet Repeat RNA-Binding Protein 1 (CELF1), two proteins with roles in mRNA splicing and RNA binding (*Figure 6E*).

To test whether ORFL147C plays an important role in viral replication, possibly via a splicing or transcriptional mechanism, an HCMV recombinant was generated in which the three most N-terminal methionine residues in ORF147C were mutated without modifying the coding sequence of UL56. The growth of ΔORFL147C virus was significantly impaired, suggesting that ORFL147C plays an

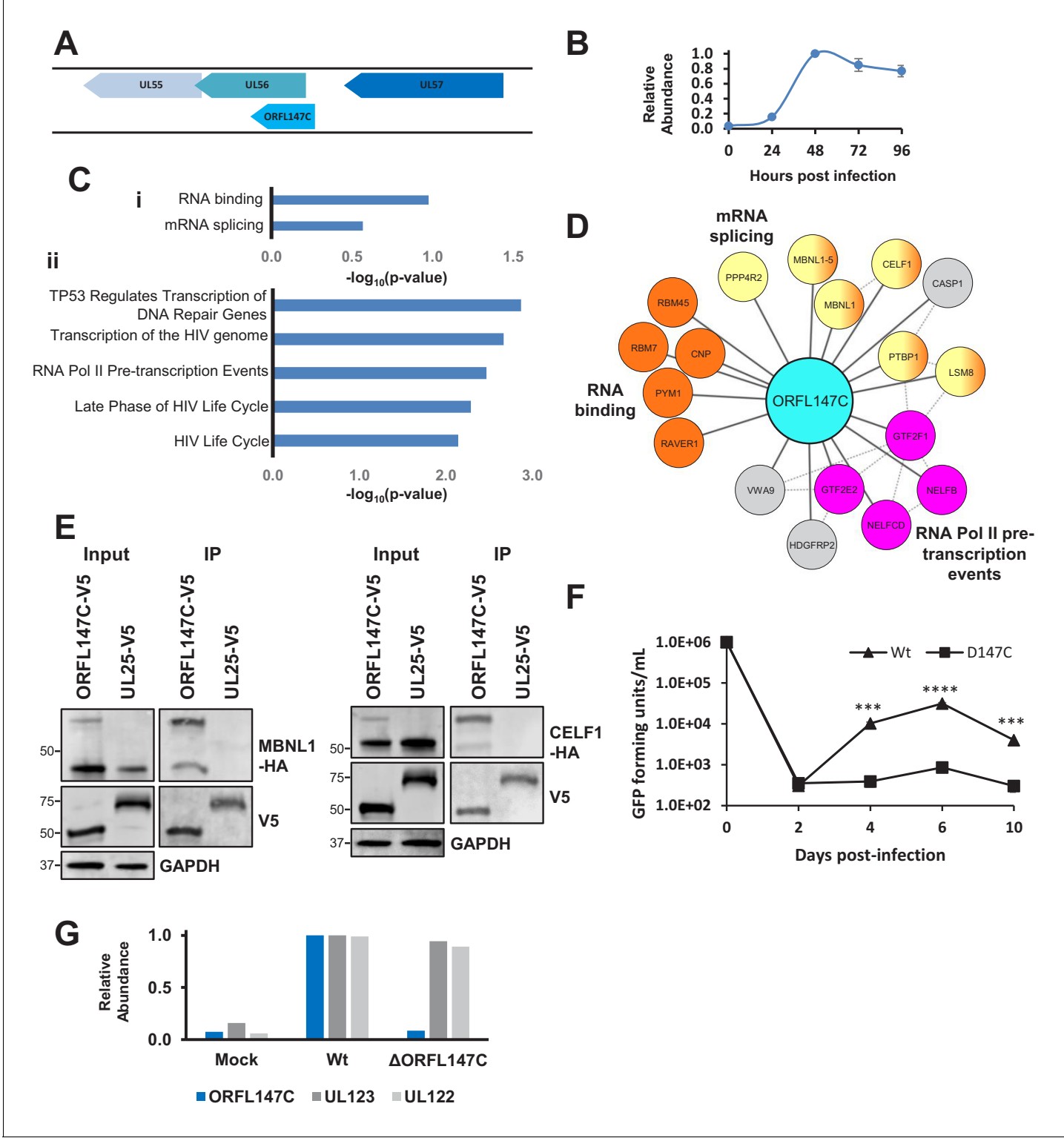

**Figure 6.** HCMV ORFL147C interactors function in RNA binding, splicing and transcription. (**A**) Diagram of the ORFL147C coding sequence and relation to neighbouring viral genes. (**B**) Expression kinetics of ORFL147C, taken from *Weekes et al. (2014)*. Data was taken from experiments WCL2 and WCL3, enabling assessment of 24, 48, 72 and 96 hr time points in biological duplicate. Error bars show range. Mean expression was normalised to a maximum of 1. (**C**) Enrichment analysis of 80 human HCIPs interacting with ORFL147C. (**i**) DAVID analysis using all human proteins as background. Benjamini-Hochberg adjusted p-values are shown. (**ii**) Reactome database analysis (*Fabregat et al., 2018*) showing results with a minimum of 4 entities per enriched pathway. Full details of interacting proteins are given in *Supplementary file 7A-B*. (**D**) A subset of HCIPs for ORFL147C (full data are

*Figure 6 continued on next page*

*Figure 6 continued*

shown in *Figure 6—figure supplement 1*). Dashed lines represent human-human interactions derived from Bioplex 2.0 as described in *Figure 2*, in addition to known interactions that had been experimentally determined or derived from curated data as part of the STRING database. (E) Validation of interaction between ORFL147C and MBNL1 and CELF1 by co-IP, conducted as described in *Figure 3*. HEK293T cells were transiently transfected with the indicated plasmids, one expressing the C-terminally V5-tagged viral protein and the other expressing C-terminally HA-tagged MBNL1 or CELF1. These proteins were detected with anti-V5 and anti-HA. GAPDH – calnexin loading control. This figure is representative of n = 1 experiment. Expected sizes: MBNL1: 33–42 kDa; CELF1: 50–55 kDa; ORFL147C: 50 kDa; UL25: 74 kDa; GAPDH: 36 kDa. (F) Growth analysis of an ORFL147C-deficient recombinant. The ORFL147C and wild-type viruses were HCMV strain Merlin recombinants in which the enhanced GFP (eGFP) gene was cloned as a 3'-terminal fusion with immediate-early gene UL36, with a self-cleaving P2A peptide releasing the reporter following synthesis. Insertion of GFP does not impede UL36 function in such recombinants (*Nightingale et al., 2018*). Cells were infected at a MOI of 1, and supernatants harvested and titred every two days. Cells were infected in biological duplicates, and each supernatant was titred in technical duplicates. Mean values are shown, and error bars represent SD. p-values for a difference between wild-type and ORFL147C-deficient virus were estimated using a two-tailed Student's t-test. ***p<0.001, ****p<0.0001. This figure is representative of n = 2 experiments. All data for this figure are also shown in *Figure 6—source data 1*. (G) ORFL147C protein is not expressed during infection with the ORFL147C-deficient recombinant (MOI = 2, 48 hr post infection). Viral protein expression was analysed using tandem mass tag-based proteomics as previously described (*Nightingale et al., 2018*). ORFL147C protein was measured at the same level as during mock infection in cells infected with the ORFL147C-deficient recombinant, attributable to noise. All data for this figure are also shown in *Figure 6—source data 2*.

The online version of this article includes the following source data and figure supplement(s) for figure 6:

**Source data 1.** Growth analysis of an ORFL147C-deficient recombinant.
**Source data 2.** Tandem mass tag-based proteomics analysis of ORFL147C protein expression.
**Figure supplement 1.** Further details of ORFL147C interactions, and construction of the ΔORFL147C virus.

important functional role during viral infection (*Figure 6F–G*). The large HCIP network for ORFL147C suggests that various mechanisms underlying this observation need to be examined; it is as yet unclear whether splicing or transcriptional effects are important.

## Discussion

In the present study, we report the largest host-pathogen interactome to date and the first comprehensive interactome map for a DNA virus in infected cells. This has suggested functions and domain associations for multiple uncharacterised or partly characterised viral proteins, in addition to providing evidence that the non-canonical HCMV proteins ORFL147C and ORFS343C may be functional. The searchable database provided details virus-virus and virus-host interactions for 162/171 HCMV proteins, and will be of significant value in future studies of HCMV and other herpesviruses.

Different herpesviruses exhibit certain common functions (*Mocarski Jr, 2007*). A previous study identified 564 human HCIPs of Kaposi's sarcoma-associated herpesvirus (KSHV) (*Davis et al., 2015*). Comparison of HCMV and KSHV interactomes revealed that baits from both viruses interacted with 176 identical human prey, including RNA Pol II, CCR4-NOT and CTLH components, and elements of the ubiquitin conjugation pathway. It will be important in future studies to determine which of these common functions are mediated by orthologous proteins, and which by distinct viral mechanisms. Conversely, certain HCMV prey did not interact with KSHV baits, including mRNA splicing machinery components (*Figure 7*). Comparisons with interactomes from additional herpesviruses when generated will help to delineate functions exhibited by all herpesvirus genera, from those more specific to individual viruses or viral subfamilies.

The combination of interactome data generated in the present study with our previous screens of protein degradation during early HCMV infection (*Nightingale et al., 2018*) identified the viral UL42 protein as a hub of degradation for multiple ubiquitin E3 ligases, and predicted novel interactions between viral baits and 29 other degraded cellular prey. More broadly, we discovered that HCMV devotes multiple proteins to interactions with the ubiquitin conjugation pathway, with 18 viral proteins interacting with two or more E3 ligases (defined in *Medvar et al., 2016*) and 51 viral proteins interacting with one or more E3 ligase. Details of such interactions can potentially identify viral mechanisms of cellular protein degradation. For example, UL25 interacted with the adaptor protein WD Repeat Domain 26 (WDR26), which can recruit substrates to the Cullin-4 RING ubiquitin ligase family (*Higa et al., 2006*). UL25 interacted with UL26, which itself interacted with 9 out of 10 members of the CTLH complex, a homologue of the yeast glucose-induced degradation-deficient machinery. This complex has inherent E3 ligase activity, but so far substrates have not been well defined

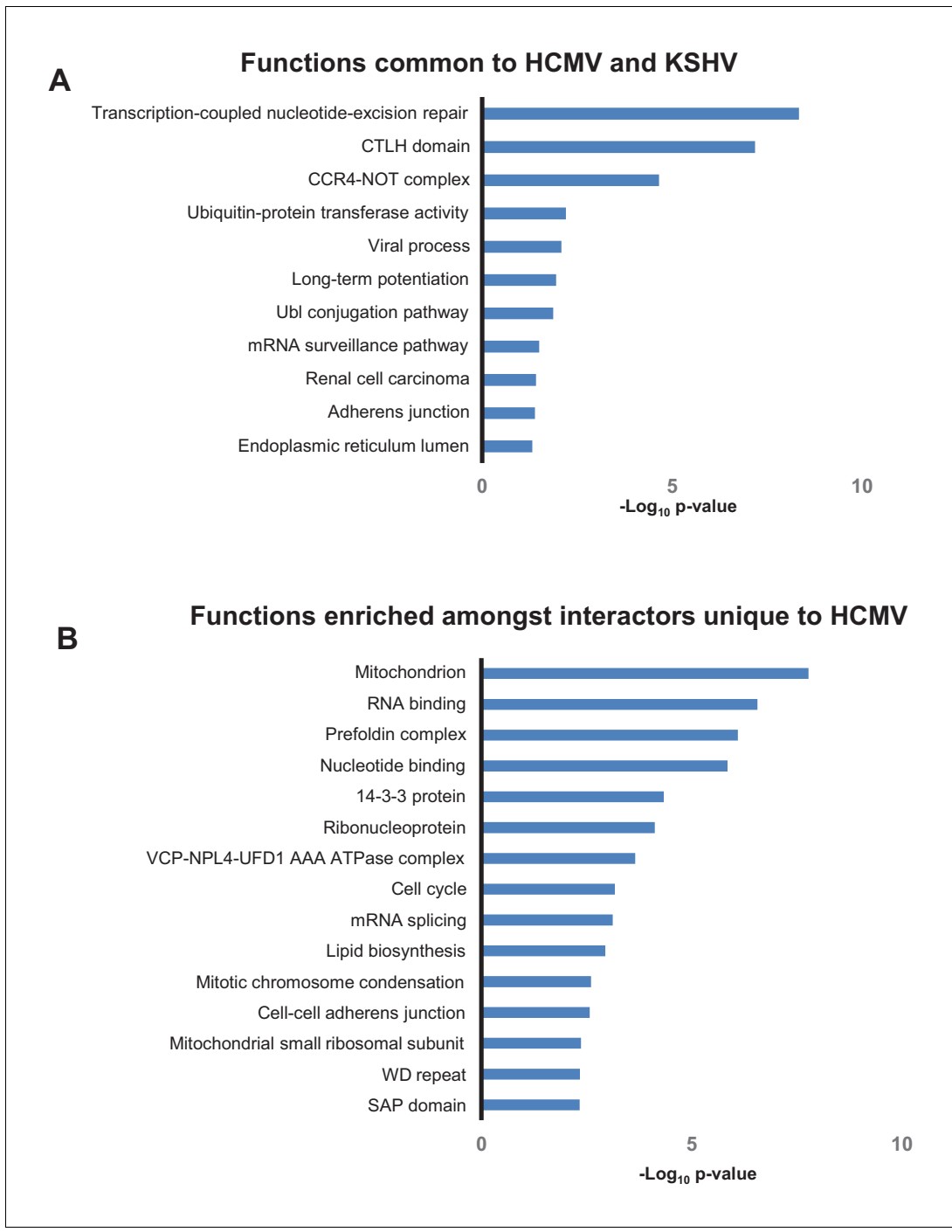

**Figure 7.** Overlap in functions targeted by different viruses. (**A**) DAVID analysis of pathway enrichment among 176 HCIPs that interacted both with HCMV baits (this study) and KSHV baits (*Davis et al., 2015*), in comparison to all human proteins as background. Benjamini-Hochberg adjusted p-values are shown for each pathway. Full details of interacting viral and host proteins are given in *Supplementary file 7A*. (**B**) DAVID analysis of pathway enrichment among HCIPs that only interacted with HCMV but not KSHV baits, in comparison to all human proteins as background. As the KSHV interactome was performed in HEK293T cells as opposed to HFFFs, the list of HCMV HCIPs was first filtered to include proteins that were clearly detectable in HEK293Ts, using the list of ~50,000 unfiltered bait-prey interactions from KSHV to indicate protein expression (*Davis et al., 2015*). Subsequently, both high confidence interacting prey of KSHV baits, and first degree interactors of these prey from the human interactome, were excluded (*Huttlin et al., 2017*), to leave a list of proteins that only interacted with HCMV.

*Figure 7 continued on next page*

*Figure 7 continued*

Benjamini-Hochberg adjusted p-values are shown for each pathway. Full details of interacting viral and host proteins are given in **Supplementary file 7B**.

(*Francis et al., 2013*; *Salemi et al., 2017*). Finally, UL26 also interacted with other ligases and scaffolds, such as Cullin three and SMAD Specific E3 Ubiquitin Protein Ligase 2 (SMURF2). Future work is likely to identify whether UL25 or UL26 prey are degraded, and which of these cellular pathways are employed.

The present study also highlights other viral 'hubs' of protein degradation. For example, HCMV UL20 was previously found to be rapidly degraded, with the suggestion it may target unidentified cellular proteins to lysosomes (*Jelcic et al., 2011*). Here, we identify candidate cellular targets. For example, UL20 interacted with Interleukin 6 Signal Transducer (IL6ST), the neonatal Fc receptor (FCGRT), Ephrin A2 (EPHA2), and Interferon Gamma Receptor 1 (IFNGR1), all of which we have previously shown are rescued from degradation by application of the lysosomal protease inhibitor Leupeptin. Interestingly, all four proteins were also rescued by targeted deletion of members of the viral US12-US21 family of paralogous genes (*Fielding et al., 2017*). This suggests that there may be cooperativity between the US12-US21 proteins and UL20, possibly with UL20 acting in a final common pathway.

All systematic interactomes of this type include false discoveries and fail to detect certain genuine interactions. However, a particular advantage of considering multiple interactions simultaneously in comparison to isolated IP-MS experiments is a much lower false discovery rate (estimated ~5%), as non-specific interacting proteins can be excluded because they are commonly identified in multiple different IPs (*Sowa et al., 2009*). The present study also identified a subset of VHCIPs by employing two distinct filtering strategies, which will assist future investigations based on our data. It is difficult to estimate a true false negative rate, since there is no gold standard for assessing true interactions, and the published literature also suffers from false discoveries. One factor that may contribute to missed identifications is the abundance of the prey protein. The present study clearly has the ability to identify some interacting proteins present at low cellular abundance, exemplified by identification of the interaction between US10 and LRFN3. LRFN3 was below the limit of detection in two unbiased quantitative proteomic studies of >8000 proteins from whole cell lysates of HFFFs (**Supplementary file 1C**). However, 36% of previously described interactions that were not identified in the present study were also unquantified in whole cell lysates (**Supplementary file 3**), suggesting that protein abundance may play a significant role in interaction discovery. Furthermore, degradation of human prey proteins during HCMV infection may also impact the limit of detection by MS. For example, although RL1 interacted with the Cullin four scaffold and two associated proteins, no other high confidence RL1 prey were identified. It will therefore be important to repeat this interactome in the presence of lysosomal and proteasomal inhibition to identify such targets. Additionally, for future investigations of our data, validation of interactions in which the prey protein has low cellular abundance as indicated in **Supplementary file 2B** may be best performed by overexpression studies as opposed to attempts to co-IP the endogenous protein.

Overexpression of each bait throughout the course of infection may have led to temporal dysregulation of the expression of other viral proteins, and may have facilitated interactions that would usually commence earlier or later than 60 hr of infection. However, as 153/153 quantified viral ORFs were expressed at 60 hr (*Weekes et al., 2014*), the observed interactions should occur at this phase of infection even if either bait or prey protein or both were not maximally expressed. Stable overexpression of the viral bait might have enabled false positive interactions. However, certain proteins endogenously expressed by HCMV are already under the control of strong promoters (*Mocarski et al., 2013*). Indeed, the abundance of certain stably expressed proteins may actually have been lower than the abundance of the same proteins expressed during HCMV infection. From our IBAQ analysis of host and viral protein abundance averaged across 24, 48 and 72 hr of HCMV infection, the most abundant viral protein (UL83) was expressed ~2.4 fold more than the most abundant host protein (Galectin-1), and the least abundant viral protein ~62 fold more than the least abundant host protein (**Supplementary file 1A**, **Supplementary file 1C**), suggesting that the range of expression of viral proteins was already shifted towards the higher end of host protein expression. Prior human interactome studies have found no correlation between bait protein expression and the

number of HCIPs (*Sowa et al., 2009*). Alternative strategies to conduct an interactome study would also suffer from potential confounding issues. For example, introduction of a tag into the viral genome prior to or after each coding sequence may facilitate expression of the bait at the same time and level as during infection with unmodified virus. However, due to the occurrence of polycistronic transcription of viral genes and overlapping viral ORFs (*Stern-Ginossar et al., 2012*), introduction of a tag may disrupt expression of neighbouring genes.

A large number of noncanonical ORFs were identified by ribosome profiling as potentially being translated (RP-ORFs, *Stern-Ginossar et al., 2012*), and 13 novel ORFs from a six-frame translation of the HCMV genome sequence were recognised as being represented in MS data (6FT-ORFs, *Nightingale et al., 2018*). However, these studies produced no evidence that any of these ORFs encode functional proteins. The present study identified three RP-ORFs and 2/13 6FT-ORFs as interactors of canonical HCMV proteins, and identified seven additional interacting 6FT-ORFs for the first time. There is thus a case for functional investigations of a modest number of additional ORFs, and initial prediction of these functions can be achieved by interaction analysis. For example, although the precise function of ORFL147C remains to be determined, we validated interactions with proteins involved in mRNA splicing including MBNL1 and CELF1. Other interactors with roles in RNA binding, such as Ribonucleotide PTB-binding 1 (RAVER1) modulates alternative splicing events. Spliced transcripts have long been recognized from HCMV at all times post infection (*Rawlinson and Barrell, 1993*), and more recently up to 100 splice junctions have been identified (*Balázs et al., 2017*; *Gatherer et al., 2011*; *Stern-Ginossar et al., 2012*).

Only three drugs are currently available to treat HCMV infection, and all suffer from significant side effects and the threat of the development of resistance. In the context of the increasing frequency of transplantation, innovative therapeutic strategies are required. The identification of key interactions in virus-virus or virus-host protein complexes may be important in this regard, since small molecule inhibitors may be able to disrupt these interactions or restore endogenous antiviral restriction by preventing host protein degradation (*Cen et al., 2010*; *Nathans et al., 2008*; *Pery et al., 2015*). To identify bait-prey pairs amenable to straightforward therapeutic interruption, it is desirable to identify factors targeted by a single viral protein, for example members of the CNOT complex by UL72. In addition to the interaction between UL72 and individual CNOT members, CNOT effector function could also be an antiviral target, for example employing inhibitors of the CNOT7 deadenylase (*Maryati et al., 2014*). Ideally, similar interactions involving several distinct pathways might be targeted simultaneously to inhibit viral replication in a way that is refractory to resistance. As an additional strategy, the recent identification of putative ligands for the viral GPCRs may facilitate approaches to targeting cytotoxins exclusively to infected cells (*Krishna et al., 2017*). These considerations illustrate the potential of the interactome data in the present study for identifying biologically important protein-protein interactions and developing antiviral therapies based on their disruption.

# Materials and methods

## Key resources table

| Reagent type | Designation | Source or reference | Identifiers | Additional information |
|---|---|---|---|---|
| Strain, strain background (HCMV) | HCMV Merlin | *Stanton et al., 2010* | RCMV1111 | |
| Strain, strain background (HCMV) | HCMV Merlin UL36-GFP deltaORFL147C | This paper | RCMV2697 | Available from Dr Michael Weekes' lab, University of Cambridge |
| Strain, strain background (HCMV) | HCMV Merlin UL36-GFP | *Nightingale et al., 2018* | RCMV2582 | |
| Strain, strain background (*Escherichia coli*) | *E. coli.* (α-Select Silver Competent Cells) | Bioline | Cat#BIO-85026 | |

*Continued on next page*

*Continued*

| Reagent type | Designation | Source or reference | Identifiers | Additional information |
|---|---|---|---|---|
| Cell line (*Homo-sapiens*) | HFFF immortalised with human telomerase (HFFF-TERT) | *McSharry et al., 2001* | | |
| Cell line (*Homo-sapiens*) | Human Embryonic Kidney 293 T cells | *Menzies et al., 2018* | ATCC Cat#CRL-3216, RRID:CVCL_0063 | |
| Antibody | Anti-V5 Agarose Affinity Gel | Sigma-Aldrich | Cat#A7345; RRID:AB_10062721 | (30 μl/mL) |
| Antibody | Mouse monoclonal anti-GAPDH | R and D Systems | Cat#MAB5718; RRID:AB_10892505 | (1:10.000) |
| Antibody | Rabbit polyclonal anti-Calnexin | LifeSpan Biosciences | Cat#LS-B6881; RRID:AB_11186721 | (1:10.000) |
| Antibody | Rabbit monoclonal anti-HA (C29F4) | Cell Signaling Technologies | Cat#3724S; RRID:AB_1549585 | (1:1000) |
| Antibody | Mouse monoclonal anti-V5 | Thermo | Cat#R960-25; RRID:AB_2556564 | (1:5000) |
| Antibody | Rabbit polyclonal anti-CNOT2 | Novus Biologicals | Cat#NBP2-56034; RRID:AB_2801658 | (1:1000) |
| Antibody | Rabbit monoclonal anti-CNOT7 | Abcam | Cat#ab195587; RRID:AB_2801659 | (1:1000) |
| Antibody | Mouse monoclonal anti-NEDD4 | R and D Systems | Cat#MAB6218; RRID:AB_10920762 | (1:1000) |
| Antibody | IRDye 680RD goat anti-mouse IgG | LI-COR | Cat#925–68070, RRID:AB_2651128 | (1:10.000) |
| Antibody | IRDye 800CW goat anti-rabbit IgG | LI-COR | Cat#925–32211, RRID:AB_2651127 | (1:10.000) |
| Antibody | IRDye 680RD goat anti-rabbit IgG | LI-COR | Cat#926–68071; RRID:AB_10956166 | (1:10.000) |
| Antibody | IRDye 800CW goat anti-mouse IgG | LI-COR | Cat#926–32210; RRID:AB_621842 | (1:10.000) |
| Antibody | Human TruStain FcX | BioLegend | Cat#422302; RRID:AB_2818986 | 1:20 |
| Recombinant DNA reagent | pHAGE-pSFFV | *Nightingale et al., 2018* | | |
| Recombinant DNA reagent | pDONR223 | *Nightingale et al., 2018* | | |
| Recombinant DNA reagent | pDONR221-MBLN1 | Harvard PlasmID | Cat#HsCD00079833 | |
| Recombinant DNA reagent | pDONR221-CUGBP1 | Harvard PlasmID | Cat#HsCD00039403 | |
| Recombinant DNA reagent | pOTB7-CUL4A | Harvard PlasmID | Cat#HsCD00325140 | |
| Recombinant DNA reagent | pCMV-SPORT6-NEDD4L | Harvard PlasmID | Cat#HsCD00337956 | |
| Recombinant DNA reagent | pENTR223-NCK1 | Harvard PlasmID | Cat#HsCD00370605 | |
| Recombinant DNA reagent | pDONR223-CNOT2 | Harvard PlasmID | Cat#HsCD00080019 | |
| Recombinant DNA reagent | pHAGE-CNOT7 | Harvard PlasmID | Cat#HsCD00453329 | |
| Recombinant DNA reagent | PHAGE-P-CMVt-N-HA Nedd4 wt | Addgene | Cat#24124 | |

*Continued on next page*

*Continued*

| Reagent type | Designation | Source or reference | Identifiers | Additional information |
|---|---|---|---|---|
| Recombinant DNA reagent | pDONR221-LRFN3 | Harvard PlasmID | Cat#HsCD00041564 | |
| Sequence-based reagent | M13-F | GENEWIZ | PCR primers | GTAAAACGACGGCCAG |
| Sequence-based reagent | M13-R | GENEWIZ | PCR primers | CAGGAAACAGCTATGAC |
| Sequence-based reagent | pHAGE-pSFFV-Seq | This paper | PCR primers | CGCGCCAGTCCTCCGATTG |
| Sequence-based reagent | GAW-CMVp-F | This paper | PCR primers | GGGACAAGTTTGTACAAAA AAGCAGCTGAAGACACCGGGACCGATC |
| Sequence-based reagent | attB2-V5-R | This paper | PCR primers | GGGGACCACTTTGTACAAGA AAGCTGGGTTTACGTAGAAT CAAGACCTAGGAGC |
| Peptide, recombinant protein | V5 Epitope Tag | Alpha Diagnostic International | Cat#SP-59199–5 | |
| Peptide, recombinant protein | Trypsin | Promega | Cat#V5111 | |
| Commercial assay or kit | BCA Protein Assay Kit | Thermo Fisher | Cat#23227 | |
| Commercial assay or kit | Micro BCA Protein Assay Kit | Thermo Fisher | Cat#23235 | |
| Commercial assay or kit | RNeasy Mini Kit | Qiagen | Cat#74104 | |
| Commercial assay or kit | Empore SPE Disks | Supelco | Cat#66883 U | |
| Commercial assay or kit | GoScript Reverse Transcriptase kit | Promega | Cat#A5001 | |
| Commercial assay or kit | Power SYBR Green PCR Master Mix | Thermo Fisher | Cat#4367659 | |
| Commercial assay or kit | Gateway BP Clonase II Enzyme Mix | Invitrogen | Cat#56481 | |
| Commercial assay or kit | Gateway LR Clonase Enzyme Mix | Invitrogen | Cat#56484 | |
| Chemical compound, drug | Dexamethasone | Sigma-Aldrich | Cat#D4902 | |
| Chemical compound, drug | DL-Dithiothreitol | Sigma-Aldrich | Cat#43815–1G | |
| Software, algorithm | 'MassPike', a Sequest-based software pipeline for quantitative proteomics. | Professor Steven Gygi's lab, Harvard Medical School, Boston, USA. | | |
| Software, algorithm | SEQUEST | *Eng et al., 1994* | | |
| Software, algorithm | DAVID software | https://david.ncifcrf.gov/ | DAVID, RRID:SCR_001881 | |
| Software, algorithm | Reactome software | https://reactome.org/ | Reactome, RRID:SCR_003485 | |
| Software, algorithm | Image Studio Lite | LI-COR | Ver. 5.2; Image Studio Lite, RRID:SCR_013715 | |
| Software, algorithm | Cytoscape | The Cytoscape Consortium | Ver 3.7.1; Cytoscape, RRID:SCR_003032 | |

*Continued on next page*

*Continued*

| Reagent type | Designation | Source or reference | Identifiers | Additional information |
|---|---|---|---|---|
| Software, algorithm | DNASTAR Lasergene - SeqBuilder | DNASTAR, Inc | Ver. 12; DNASTAR: Lasergene Core Suite, RRID:SCR_000291 | |
| Software, algorithm | FlowJo | FlowJo | Ver. 10; FlowJo, RRID:SCR_008520 | |
| Software, algorithm | CompPass | *Sowa et al., 2009* | | |
| Software, algorithm | CompPass Plus | *Huttlin et al., 2015* | | |
| Other | Orbitrap Fusion Mass Spectrometer | Thermo Fisher Scientific | Cat#IQLAAEGAAP FADBMBCX | Instrument |
| Other | Orbitrap Fusion Lumos Mass Spectrometer | Thermo Fisher Scientific | Cat#IQLAAEGAAP FADBMBHQ | Instrument |
| Other | Raw Mass Spectrometry Data Files | This paper | ProteomeXchange Consortium via the PRIDE partner repository with dataset identifier PXD014845. | Raw data |

## Cells and cell culture

Human fetal foreskin fibroblast cells immortalised with human telomerase (HFFF-TERTs, male) and HEK293T cells (female) were grown in Dulbecco's modified Eagle's medium (DMEM) supplemented with foetal bovine serum (FBS: 10% v/v), and 100 IU/ml penicillin/0.1 mg/ml streptomycin (DMEM/FBS/PS) at 37°C in 5% v/v $CO_2$. HFFF-TERTs have been tested at regular intervals since isolation to confirm that human leukocyte antigen (HLA) and MHC Class I Polypeptide-Related Sequence A (MICA) genotypes, cell morphology and antibiotic resistance are unchanged. In addition, HCMV strain Merlin grows only in human fibroblast cells (dermal or foreskin in origin), further reducing the possibility that they have been contaminated with another cell type. HEK293T cells were obtained as a gift from Professor Paul Lehner and had been authenticated by Short Tandem Repeat profiling (*Menzies et al., 2018*). All cells were confirmed to be mycoplasma-negative (Lonza MycoAlert).

## Viruses

The genome sequence of HCMV strain Merlin is designated the reference for HCMV by the National Center for Biotechnology Information, and was originally sequenced after three passages in human fibroblast cells (*Dolan et al., 2004*). A recombinant version (RCMV1111) of this strain was derived by transfection of a sequenced BAC clone (*Stanton et al., 2010*). RCMV1111 contains point mutations in two genes (RL13 and UL128) that enhance replication in fibroblasts (*Stanton et al., 2010*).

HCMV expressing rGFP from a P2A self-cleaving peptide at the 3'-end of the UL36 coding region (RCMV2582) was generated by recombineering the strain Merlin BAC as described previously (*Stanton et al., 2010*). An ORFL147C mutant (RCMV2697) was generated by recombineering RCMV2582. Substitutions were introduced into three in-frame ATG codons at or near the 5'-end of ORFL147C, in such a way that the coding potential of UL56, with which ORFL147C overlaps extensively in another reading frame, was unaffected. Whole-genome consensus sequences of passage 2 of all recombinant viruses were derived using the Illumina platform as described previously (*Fielding et al., 2014*).

Viral stocks were prepared from HFFF-TERTs as described previously (*Stanton et al., 2007*). When complete cytopathic effect was observed, cell culture supernatants were centrifuged to remove cell debris and then centrifuged at 22,000 × g for 2 hr to pellet cell-free virus. The virus was resuspended in fresh DMEM, and residual debris was removed by centrifugation at 16,000 x g for 1 min. In total, 17 stocks of RCMV1111 were required for this project. To ensure identical infection conditions between every batch of viral infections, each stock was divided into 25 aliquots. For each batch of infections, one aliquot of each stock was thawed, and then all 17 aliquots were combined and mixed prior to infection.

## Plasmid construction

For the majority of HCMV genes, a library of recombinant adenovirus vectors (RAds) was used to generate lentiviral constructs. Each template expresses a C-terminally V5-tagged gene under the control of the HCMV major immediate early promoter, with a 6 bp linker region between the end of the gene and the tag. Of 169 genes cloned into RAds, expression was confirmed for 160 by a combination of IB (152 genes) and immunofluorescence (155 genes). The codon usage of US14, US17 and UL74 was optimized for expression (*Supplementary file 1D*) (*Seirafian, 2012*). To amplify genes from the RAds, primers were designed to recognise the 3' end of the HCMV promoter (forward 'GAW-CMVp-F') and the 3' end of the V5 tag (reverse 'attB2-V5-R'). Both primers had flanking Gateway attB sequences (*Supplementary file 1E*, Key Resources Table).

For HCMV genes amplified from the RCMV1111 BAC, primers were designed to recognize the 3' end and the 5' end of each gene (*Supplementary file 1E*). In addition to the gene-specific sequence, the reverse primer also contained a 6 bp linker region, followed by the coding sequence for the V5 tag and a stop codon. Both primers had flanking Gateway attB sequences.

A subset of HCMV genes was synthesized as double-stranded DNA fragments (gBlocks, Integrated DNA Technologies, detailed in the *Supplementary file 1E* 'Template' column). Each fragment comprised the viral gene (without a stop codon), succeeded by a 6 bp linker region, the coding sequence for the V5 tag then the stop codon. The fragments had flanking Gateway attB sequences. The sequences of all primers and HCMV genes used in this study are shown in *Supplementary file 1D-E* and the Key Resources Table.

Two control vectors were additionally employed. 'GAW Control' contains a short DNA sequence (produced by a random DNA sequence generator) flanked by Gateway attB sequences. Complementary oligonucleotides (*Supplementary file 1E*) were annealed to generate a double-stranded DNA fragment, which was then inserted into pDONR223 by gateway recombination. A second control vector coding for GFP was cloned from the adenoviral template library as described above. Neither the 'GAW Control' nor GFP were tagged with V5.

For HA-tagged human genes (*Figures 3–6*), primers were designed to recognise the 3' end and 5' end of each gene (*Supplementary file 1F*). In addition to the gene-specific sequence, the reverse primer also contained a 6 bp linker region, followed by the coding sequence for an HA tag and a stop codon. Both primers had flanking Gateway attB sequences.

PCR employed PfuUltra II Fusion HS DNA polymerase (Agilent). Constructs were subsequently cloned into the pDONR223 entry vector, then into the lentiviral destination vector pHAGE-pSFFV using the Gateway system (Thermo Scientific). pHAGE-pSFFV has a spleen focus-forming virus (SFFV) promoter replacing the HCMV promoter in pHAGE-pCMV to prevent promoter inactivation during HCMV infection (*Nightingale et al., 2018*). For UL48, which is 6.7 kbp long, it was not possible to express the whole construct via lentiviral transduction alone, probably due to inefficient transduction. UL48 contains a predicted α-helix from residues 540–1500, but no predicted secondary structure between residues 1501–1509. The gene was therefore divided into two segments, one of 4.5 kbp (1–1504 aa) terminating in a stop codon, and one of 2.2 kbp (1505–2241 aa), with an additional start codon. Both segments were stably expressed in different cell lines, and HCMV-infected cellular lysates were combined prior to IP. Full sequencing of all genes was conducted in the pDONR223 vector using standard primers and additional internal primers as required (Key Resources Table). All pHAGE-pSFFV vectors underwent sequencing of the first ~700 nucleotides from the 3' end of the SFFV promoter to verify that the viral construct had recombined correctly.

## Stable cell line production

Lentiviral particles were generated through transfection of HEK293T cells with the lentiviral transfer vector and four helper plasmids (VSVG, TAT1B, MGPM2, CMV-Rev1B), using TransIT-293 transfection reagent (Mirus) according to the manufacturer's recommendations (*Nightingale et al., 2018*). Viral supernatant was harvested 48 hr post-transfection and cell debris was removed with a 0.22 μm filter. To facilitate stable, constitutive expression of the viral transgene, target cells were transduced for 48 hr and then subjected to antibiotic selection for two weeks.

## Immunoblotting to confirm viral bait expression

Lysates for each HFFF-TERT cell line expressing a viral bait were tested for transgene expression by IB for the V5 tag. Cells were lysed with RIPA buffer (Cell Signaling) containing Complete Protease Inhibitor Cocktail (Roche) and clarified by centrifugation at 16,000 x g for 10 min. Protein concentration was measured by BCA (Pierce) using the manufacturer's protocol. Lysates were reduced with 6X Protein Loading Dye (375 mM Tris-HCl pH 6.8, 12% w/v sodium dodecyl sulphate (SDS), 30% v/v glycerol, 0.6 M dithiothreitol (DTT), 0.06% w/v bromophenol blue) for 5 min at 95°C. 20 μg of protein for each sample was separated by polyacrylamide gel electrophoresis (PAGE) using 4–15% TGX Precast Protein Gels (Bio-rad), then transferred to polyvinylidene difluoride (PVDF) membranes using Trans-Blot Systems (Bio-rad). The following primary antibodies were used: anti-V5 (MA5-15253, Thermo) and anti-Calnexin (CANX, LS-B6881, LifeSpan BioSciences). Secondary antibodies were IRDye 680RD goat anti-rabbit (926–68071, LI-COR) and IRDye 800CW goat anti-mouse (926–32210, LI-COR). Fluorescent signals were detected using a LI-COR Odyssey, and images were processed using Image Studio Lite (LI-COR).

Where viral baits could not be detected by IB, IP-MS was used with uninfected cellular lysates as described below in 'IP and protein digestion for proteomic experiments'. Where a bait could not be detected by IP-MS, RT-qPCR was used as described below.

## RT-qPCR to confirm viral bait expression

Total RNA from a subset of HFFF-TERT lines expressing viral transgenes was extracted using an RNeasy Mini Kit (Qiagen). cDNA was synthesized using GoScript Reverse Transcriptase (Promega), followed by RT-qPCR using Fast SYBR Green Master Mix (Applied Biosystems) and 7500 Fast and 7500 Real-Time PCR Systems (Applied Biosystems). Primers targeting HCMV genes or GAPDH (as an internal control) are shown in *Supplementary file 1E*. The PCR program started with activation at 95°C for 2 min, followed by 40 cycles of denaturation at 95°C for 5 s and annealing/extension at 60°C for 30 s. The amplification products were then separated by agarose gel electrophoresis, purified (QIAquick Gel Extraction, Qiagen) and sequenced to confirm viral bait expression. For UL146 and UL148D, this procedure failed to generate sequenceable amplicons, and UL136 failed to generate any PCR product despite the use of primers that recognized both a short and full-sized amplicon (*Supplementary file 1E*). For UL146 and UL148D, whole gene amplicons (189–363 bp) were generated by PCR with PfuUltra II Fusion HS DNA Polymerase (Agilent), according to the manufacturer's recommendations. Sequencing of the amplified product confirmed expression of the correct gene in each case.

## Virus infections for IP-MS proteomic experiments

Each batch of viral infections included eight cell lines stably expressing different viral baits in duplicate. For each cell line, $6 \times 10^6$ cells were plated in DMEM/FBS/PS in each of two 150 cm$^2$ dishes. After 24 hr, the medium was changed to DMEM lacking FBS but with 4 μg/ml dexamethasone, as this approach has been shown to improve infection efficiency (*Tanaka et al., 1984*). After 24 hr, the medium was changed to DMEM containing the requisite volume of HCMV strain Merlin stock to achieve MOI 2. Cells were gently rocked for 2 hr, and then the medium was changed to DMEM/FBS/PS and cells were incubated for a further 58 hr.

## IP and protein digestion for IP-MS proteomic experiments

Cells were harvested in one of two lysis buffers in order to best solubilise each bait protein and preserve protein-protein interactions. For soluble and single-pass transmembrane (TM) baits, cells were lysed in 50 mM Tris-HCl pH 7.5, 300 mM NaCl, 0.5% v/v NP40, 1 mM DTT and Roche protease inhibitor cocktail. Baits with two or more TM domains were solubilized in 1% w/v digitonin (Merck Millipore) in TBS (Sigma) and Roche protease inhibitor cocktail. Transmembrane predictions were derived from Uniprot (www.uniprot.org) for canonical HCMV proteins, and generated using TMHMM for the two novel proteins (*Krogh et al., 2001*). Samples were tumbled for 15 min at 4°C and then centrifuged at 16,100 g for 15 min at 4°C. Lysates were then clarified by filtration through a 0.7 μm filter and incubated for 3 hr with immobilised mouse monoclonal anti-V5 agarose resin (Sigma). Duplicate samples were combined for resin washes. Samples lysed in NP40-containing buffer were washed seven times with lysis buffer, followed by seven PBS pH 7.4 washes. Samples lysed in

digitonin-containing buffer were washed once with lysis buffer, twice with 0.2% (w/v) digitonin in TBS and then once with TBS. Subsequently, proteins bound to the anti-V5 resin were eluted twice by adding 200 µl of 250 µg/ml V5 peptide (Alpha Diagnostic International) in PBS at 37°C for 30 min with agitation. Finally, proteins were precipitated with 20% trichloroacetic acid (TCA), washed once with 10% TCA, washed three times with cold acetone and dried to completion using a centrifugal evaporator. Samples were resuspended in digestion buffer (50 mM Tris-HCl pH 8.5, 10% acetonitrile (AcN), 1 mM DTT, 10 ug/ml Trypsin) and incubated overnight at 37°C with agitation. The reaction was quenched with 50% formic acid (FA), subjected to C18 solid-phase extraction, and vacuum-centrifuged to complete dryness. Samples were reconstituted in 4% acetonitrile/5% formic acid and divided into technical duplicates prior to LC-MS/MS on an Orbitrap Lumos. To minimise variability in sample preparation, all samples were lysed with aliquots from the same batch of lysis buffer. Similarly, several batches of the anti-V5 agarose resin used for immunoprecipitation were pooled and this pool was used for all samples. In addition, all V5 peptide used for protein elution was derived from the same manufacturer's batch, and all protein digests were performed with aliquots from the same stock of digestion buffer.

## LC-MS/MS for IP-MS experiments

Peptides for each sample were analysed in technical duplicate, with the run order reversed from one batch of replicate analyses to the next to ensure that any carry-over was different in each case. Two washes were used between each sample to further minimise carry-over (i.e. Run 1: Sample A, wash, wash, Sample B, wash, wash, Sample C...; Run 2: ...Sample C, wash, wash, Sample B, wash, wash, Sample A). Individual batches included 16–22 samples. To ensure consistent performance by the mass spectrometer between batches, an identical aliquot of a control IP of uninfected cells stably expressing the viral UL123 gene with a C-terminal V5 tag was included with each batch. The number of peptides in total, from the bait and from known UL123 prey were very similar between batches.

The major reason for pooling the biological replicates and analysing samples in technical duplicate was to solve certain technical issues. Specifically, due to the potential for carry-over of peptides between adjacent injections of different IP samples, it was necessary to use consistency of detection of prey as a measure of confidence in bait-prey interaction. To electronically filter out carry-over contaminants, an entropy score (described below in 'Data analysis') compared the number of peptide-spectrum matches (PSM) between technical replicate injections and eliminated prey that were not detected consistently. In addition, this form of replicate analysis also enabled false positive interactions to be minimised since the same random incorrect interaction was unlikely to appear twice in two different runs. It was therefore important that replicate injection material was as similar as possible to ensure that this filter was efficacious. These issues are well understood and examined in *Huttlin et al. (2015)*; the CompPass algorithm (described below) was developed based on this specific protocol.

To directly examine biological variability, six of the IP-MS experiments were re-run, with independent analysis of each replicate. There was very good correlation between the number of PSM from each identified prey protein both between biological as well as between technical replicates (*Figure 1—figure supplement 1C–D*).

Mass spectrometry data were acquired using an Orbitrap Fusion Lumos. An Ultimate 3000 RSLC nano UHPLC equipped with a 300 µm ID x 5 mm Acclaim PepMap µ-Precolumn (Thermo Fisher Scientific) and a 75 µm ID x 75 cm 2 µm particle Acclaim PepMap RSLC analytical column was used.

Loading solvent was 0.1% v/v FA, and the analytical solvents were (A) 0.1% v/v FA and (B) 80% v/v AcN + 0.1% v/v FA. All separations were carried out at 55°C. Samples were loaded at 5 µl/min for 5 min in loading solvent before beginning the analytical gradient. The following gradient was used: 3–7% B over 3 min then 7–37% B over 54 min followed by a 4 min wash in 95% B and equilibration in 3% B for 15 min. The following settings were used: MS1, 350–1500 Thompsons (Th), 120,000 resolution, $2 \times 10^5$ automatic gain control (AGC) target, 50 ms maximum injection time. MS2, quadrupole isolation at an isolation width of m/z 0.7, higher-energy collisional dissociation (HCD) fragmentation (normalised collision energy (NCE) 34) with fragment ions scanning in the ion trap from m/z 120, $1 \times 10^4$ AGC target, 250 ms maximum injection time, with ions accumulated for all parallelisable times. The method excluded undetermined and very high charge states ($\geq$25+). Dynamic exclusion was set to + /- 10 ppm for 25 s. MS2 fragmentation was trigged on precursors 5 $\times 10^3$ counts and above. Two 45 min washes were included between every affinity purification-mass

spectrometry (AP-MS) analysis, to minimise carry-over between samples. 1 µl transport solution (0.1% v/v trifluoroacetic acid) was injected, over the following gradient: 3–40% B over 29 min followed by a 3 min wash at 95% B and equilibration at 3% B for 10 min.

## Confirmation of ORFL147C deletion in Δorfl147c recombinant virus

For *Figure 6G*, HFFF-TERT cells were infected as otherwise described in 'Virus infections for IP-MS proteomic experiments' with the following modifications: $1.5 \times 10^5$ cells seeded per well of a 12-well plate for a total of 48 hr infection. A total infection duration of 48 hr was selected as ORFL147C expression peaks at this time (*Figure 6B*).

As described in *Nightingale et al. (2018)* and briefly recapitulated here, cells were washed with PBS, lysed with 6M Guanidine/50 mM HEPES pH 8.5, scraped, vortexed extensively and sonicated, and debris was removed by centrifugation. Proteins were reduced using DTT, and cysteines alkylated with iodoacetamide, which was quenched with DTT. Samples were diluted with HEPES pH 8.5 to 1.5 M Guanidine followed by digestion at room temperature for 3 hr with LysC protease at a 1:100 protease-to-protein ratio. Samples were further diluted with 200 mM HEPES pH 8.5 to 0.5 M Guanidine. Trypsin was then added at a 1:100 protease-to-protein ratio followed by overnight incubation at 37˚C. The reaction was quenched with 5% FA, then centrifuged at 21,000 g for 10 min to remove undigested protein. Peptides were subjected to C18 solid-phase extraction (SPE, Sep-Pak, Waters) and vacuum-centrifuged to near-dryness.

Desalted peptides were dissolved in 200 mM HEPES pH 8.5, and peptide concentration was measured by Micro BCA (Thermo Fisher Scientific), then 15 µg of peptide were labeled with TMT reagent (mock - 126; wild-type – 127N; ΔORFL147C – 128N). After 1 hr, the reaction was quenched and the samples were combined 1:1:1. The combined sample was vacuum-centrifuged to near dryness and subjected to C18 SPE (Sep-Pak, Waters). Six fractions generated using high pH reversed phase fractionation as previously described (*Nightingale et al., 2018*) were analysed to increase the overall number of peptides quantified.

Mass spectrometry data were acquired using an Orbitrap Lumos as previously described (*Nightingale et al., 2018*). An Ultimate 3000 RSLC nano UHPLC equipped with a 300 µm internal diameter (ID) x 5 mm Acclaim PepMap µ-Precolumn (Thermo Fisher Scientific) and a 75 µm ID x 50 cm 2.1 µm particle Acclaim PepMap RSLC analytical column was used. Loading solvent was 0.1% FA, analytical solvent A: 0.1% FA and B: 80% AcN + 0.1% FA. All separations were carried out at 55˚C. Samples were loaded at 5 µL/minute for 5 min in loading solvent before beginning the analytical gradient. The following gradient was used: 3–7% B over 3 min, 7–37% B over 173 min, followed by a 4 min wash at 95% B and equilibration at 3% B for 15 min. Each analysis used a MultiNotch MS3-based TMT method (*McAlister et al., 2014*). The following settings were used: MS1: 380–1500 Th, 120,000 Resolution, $2 \times 10^5$ automatic gain control (AGC) target, 50 ms maximum injection time. MS2: Quadrupole isolation at an isolation width of m/z 0.7, collision-induced dissociation (CID) fragmentation (normalised collision energy (NCE) 35) with ion trap scanning in turbo mode from m/z 120, $1.5 \times 10^4$ AGC target, 120 ms maximum injection time. MS3: In Synchronous Precursor Selection mode the top 6 MS2 ions were selected for higher-energy collisional dissociation (HCD) fragmentation (NCE 65) and scanned in the Orbitrap at 60,000 resolution with an AGC target of $1 \times 10^5$ and a maximum accumulation time of 150 ms. Ions were not accumulated for all parallelisable time. The entire MS/MS/MS cycle had a target time of 3 s. Dynamic exclusion was set to + /- 10 ppm for 70 s. MS2 fragmentation was trigged on precursors $5 \times 10^3$ counts and above. Data analysis is discussed below.

## Transient transfection

$7.5 \times 10^5$ HEK293T cells were plated in each well of a 6-well dish 24 hr prior to transfection. A total of 2.5 µg plasmid DNA was transfected using TransIT-293 transfection reagent (Mirus) according to the manufacturer's recommendations. Cell lysates were harvested 48 hr post-transfection as detailed below.

## Site-directed mutagenesis

A method based on PCR overlap extension was used to generate point mutations in the coding sequence of NCK1. Primer sequences spanning the target region were generated incorporating the

desired sequence changes in both forward and reverse orientations. These, along with primers that would anneal at the 5' and 3' ends of the full-length NCK1 coding sequence (NCK1F and NCK1R, respectively) were used to amplify two fragments of NCK1, each incorporating the point mutation. Fragments were purified and assembled into a full-length mutant NCK1 coding sequence by a second round of PCR using only NCK1F and NCK1R. The product was then purified and subcloned as described above. A truncation mutant of UL25 was generated by a single round of PCR using an appropriate internal primer.

## Co-IP

HEK293T cells were used in all experiments. Cells were harvested and lysed in MCLB (50 mM Tris-HCl pH 7.5, 300 mM NaCl, 0.5% v/v NP40, 1 mM DTT and Roche protease inhibitor cocktail). Samples were tumbled for 15 min at 4°C and then centrifuged at 16,100 g for 15 min at 4°C. Lysates were then clarified by filtration through a 0.7 μm filter and incubated for 3 hr with immobilised mouse monoclonal anti-V5 agarose resin. Samples were washed three times with lysis buffer, followed by two PBS pH 7.4 washes. Subsequently, proteins bound to the anti-V5 resin were eluted once by adding 40 μl of 2.5 mg/ml V5 peptide (Alpha Diagnostic International) in PBS at 37°C for 30 min with agitation. Lysates were reduced with 6X Protein Loading Dye (375 mM Tris-HCl pH 6.8, 12% w/v SDS, 30% v/v glycerol, 0.6 M DTT, 0.06% w/v bromophenol blue) for 5 min at 95°C. 50 μg of protein for each sample was separated by PAGE using 4–15% TGX Precast Protein Gels (Bio-rad), and then transferred to PVDF membranes using Trans-Blot Systems (Bio-rad). The following primary antibodies were used: anti-Calnexin (CANX, LS-B6881, LifeSpan BioSciences), anti-GAPDH (MAB5718, R and D Systems), anti-V5 (MA5-15253, Thermo), anti-HA (C29F4, Cell Signaling), anti-CNOT2 (NBP2-56034, Novus), anti-CNOT7 (ab195587, Abcam), anti-NEDD4 (MAB6218, R and D Systems). Secondary antibodies were IRDye 680RD goat anti-mouse (925–68070, LI-COR), IRDye 680RD goat anti-rabbit (926–68071, LI-COR), IRDye 800CW goat anti-mouse (926–32210, LI-COR) and IRDye 800CW goat anti-rabbit (925–32211, LI-COR). Fluorescent signals were detected using a LI-COR Odyssey, and images were processed using Image Studio Lite (LI-COR).

## Viral growth curve analysis

For each virus stock, $1 \times 10^6$ HFFF TERTs were seeded in duplicate T25 flasks in DMEM/FBS/PS. After 24 hr, the medium was changed to 1 ml DMEM containing the requisite volume of HCMV strain Merlin stock to achieve MOI 1, and the cells were rocked gently. After adsorption for 2 hr at 37°C, unbound virus was removed by washing with DMEM. Cells were then overlaid with 5 ml DMEM/FBS/PS. Every 48 hr, all medium was removed and replaced. 1 ml aliquots of removed media were retained for titration in fibroblasts. Cells in these aliquots that had detached from the monolayer were pelleted by centrifugation at 400 x g for 10 min at 18°C and discarded. Prior to titration, all supernatants were stored at −70°C.

Titrations of cell-free virus were performed simultaneously in fibroblasts by flow cytometry, using UL36-GFP expression as a marker to calculate the percentage of infection at 24 hr PI.

## Data analysis

In the following description, the first report in the literature for each relevant algorithm is listed. Mass spectra were processed using MassPike, which is a Sequest-based software pipeline for quantitative proteomics, through a collaborative arrangement with Professor Steven Gygi's laboratory at Harvard Medical School. MS spectra were converted to mzXML using an extractor built upon Thermo Fisher's RAW File Reader library (version 4.0.26). This software is a component of the MassPike software platform and is licensed by Harvard Medical School.

A combined database was constructed as described in *Nightingale et al. (2018)* from (a) the human Uniprot database (accessed 26 January 2017), (b) the HCMV strain Merlin Uniprot database, (c) all additional non-canonical human cytomegalovirus proteins described by *Stern-Ginossar et al. (2012)*, (d) a six-frame translation of the HCMV strain Merlin genome filtered to include all ORFs of ≥8 codons (delimited by stop codons rather than requiring an initiating ATG codon), and (e) common contaminants such as porcine trypsin and endoproteinase LysC. ORFs from the six-frame translation (6FT-ORFs) were named as follows: 6FT_Frame_ORFnumber_length, where Frame is numbered 1–6, and length is in amino acid residues. The combined database was concatenated with

a reverse database composed of all protein sequences in reversed order. Searches were performed using a 20 ppm precursor ion tolerance (*Haas et al., 2006*). Product ion tolerance was set to 0.03 Th. Oxidation of methionine residues (15.99492 Da) was set as a variable modification. Peptides were assumed to be fully tryptic with up to two missed cleavages.

To control the fraction of erroneous protein identifications, a target-decoy strategy was employed (*Elias and Gygi, 2007*; *Elias and Gygi, 2010*). Peptide spectral matches (PSMs) were filtered to an initial peptide-level false discovery rate (FDR) of 1% with subsequent filtering to attain a final protein-level FDR of 1% (*Kim et al., 2011*; *Wu et al., 2011*). PSM filtering was performed using linear discriminant analysis as described previously (*Huttlin et al., 2010*). Filtering was implemented in R using the linear discriminant analysis (LDA) function in the package MASS (https://cran.r-project.org/web/packages/MASS/). This distinguishes correct from incorrect peptide identifications in a manner analogous to the widely used Percolator algorithm (*Käll et al., 2007*), although employing a distinct machine-learning algorithm. The following parameters were considered: XCorr, ΔCn, missed cleavages, peptide length, charge state, and precursor mass accuracy. Peptides shorter than seven amino acids in length or with XCorr less than 1.0 were excluded prior to LDA filtering. Peptides were then assembled into proteins and the resulting protein IDs were scored probalistically and filtered to a 1% protein-level FDR.

For MS3-based TMT, as previously described (*Nightingale et al., 2018*), TMT tags on lysine residues and peptide N termini (229.162932 Da) and carbamidomethylation of cysteine residues (57.02146 Da) were included as static modifications. Proteins were quantified by summing TMT reporter ion counts across all matching peptide-spectral matches using 'MassPike', as described previously (*McAlister et al., 2014*). Briefly, a 0.003 Th window around the theoretical m/z of each reporter ion (126, 127 n, 128 n) was scanned for ions, and the maximum intensity nearest to the theoretical m/z was used. An isolation specificity filter with a cutoff of 50% was employed to minimise peptide co-isolation (*McAlister et al., 2014*). Peptide-spectral matches with poor quality MS3 spectra (more than 3 TMT channels missing and/or a combined S:N ratio of less than 100 across all TMT reporter ions) or no MS3 spectra at all were excluded from quantitation. Peptides meeting the stated criteria for reliable quantitation were then summed by parent protein, in effect weighting the contributions of individual peptides to the total protein signal based on their individual TMT reporter ion yields. Protein quantitation values were exported for further analysis in Excel.

For protein quantitation, reverse and contaminant proteins were removed, then each reporter ion channel was summed across all quantified proteins and normalised assuming equal protein loading across all channels. For further analysis and display in *Figure 6G*, fractional TMT signals were used (i.e. reporting the fraction of maximal signal observed for each protein in each TMT channel, rather than the absolute normalised signal intensity). This effectively corrected for differences in the numbers of peptides observed per protein.

## Interactor identification with CompPASS

To identify HCIPs for each bait, replicate pairs were combined to attain a summary of proteins identified in both runs. Peptides within replicates were reassembled into proteins following the principles of parsimony (*Huttlin et al., 2010*). Where all PSMs from a given HCMV protein could be explained either by a canonical gene or a non-canonical ORF, the canonical gene was picked in preference. In four cases (UL24/ORFL71C_(UL24), UL31/ORFL87W_(UL31), UL150A/ORFL321W, UL44/ORFL112C_(UL44)), PSMs assigned to a non-canonical ORF were a mixture of peptides from the canonical protein and the ORF. This occurred where the ORF was a 5'-terminal extension of the canonical protein (thus meaning that the smallest set of proteins necessary to account for all observed peptides included the ORFs alone). In these cases, the peptides corresponding to the canonical protein were separated from those unique to the ORF, generating two separate entries.

CompPASS scoring was performed as described previously (*Huttlin et al., 2015*), in two analyses that were subsequently combined, one for NP40-based IPs and the other for digitonin IPs. These data were treated separately to better model detergent-specific differences in IP-MS background. Data reported for each protein in every IP in the dataset include: (a) the number of peptide spectrum matches (PSMs) averaged between technical replicates; (b) an entropy score, which compares the number of PSM between replicates to eliminate proteins that are not detected consistently; (c) a z-score, calculated in comparison to the average and standard deviation of PSMs observed across all IPs; and (d) an NWD score, which reflects (i) how frequently this protein was detected and (ii)

whether it was detected reproducibly. NWD scores were calculated as described in *Behrends et al. (2010)* using the fraction of runs in which a protein was observed, the observed number of PSMs, the average and standard deviation of PSMs observed for that protein across all IPs, and the number of replicates (1 or 2) containing the protein of interest. NWD scores were normalised so that the top 2% earned scores $\geq$ 1.0. For NP40-based IPs, the top 2% of z-scores were >6.676, and for digitonin-based IPs they were >4.329.

As the set of digitonin-based IPs was necessarily smaller than the set of NP40-based IPs (18 compared to 153 viral genes examined respectively), additional control IPs were included. Biological duplicates of cells transduced with empty vector controls ('GAW control'), and biological duplicates of cells transduced with a vector encoding green fluorescent protein (GFP) were included in the digitonin set. A single replicate of the GAW control was included in the NP40 set. These controls had the effect of increasing the number of IPs that identified non-specific interacting proteins, thus decreasing NWD and z-scores for these proteins. Mass spectrometry RAW files from control UL123 IPs included to ensure batch-to-batch consistency were not included in the final data analysis, to avoid modification of NWD and z-scores for the infected UL123-expressing sample.

Following CompPASS analysis, a series of filters were applied to remove inconsistent and low-confidence protein identifications across all IPs and minimise both false protein identifications and associations. These included: (a) a minimum PSM score of 1.5 (i.e. a minimum of 3 peptides per protein across both replicates); (b) a minimum entropy score of 0.75; (c) a top 2% NWD or z-score. Previous studies have estimated a 5% false discovery rate when employing a similar strategy with a top 2% NWD score (*Sowa et al., 2009*). Interactions passing these criteria are shown in *Supplementary file 2B*, and used in all subsequent analyses throughout this work. As found in prior human interactome investigations, certain known interactions fell just below the stringent top 2% NWD or z-score cutoffs. Proteins were therefore also included with top 5% NWD or z-scores (>0.434 and >3.688, respectively), if they had been reported to interact with the bait in a prior study (*Gallegos et al., 2016*). For protein UL133 (2 TM regions), an initial digitonin-based AP-MS analysis failed to generate any interactors after filtering. This IP was repeated using the NP40-based lysis buffer.

For added stringency with baits solubilised in NP40, the supervised learning algorithm CompPass Plus was employed. This identifies HCIPs whilst minimising both false positive protein IDs and background proteins as described previously (*Huttlin et al., 2017*). The CompPass Plus model was trained using known HCMV protein interactions drawn from BioGRID, IntAct, Uniprot, MINT, and Virus Mentha; incorrect protein IDs were modeled using the target-decoy method. Results reported from this algorithm include p(Interactor), the probability that a given prey is a specific interactor. We considered interactions that passed CompPass filters, had p(Interactor) values of >0.75 from CompPass Plus and in which the prey was identified by at least two unique peptides as a VHCIP. These are also indicated in *Supplementary file 2B*. Cytoscape ver 3.7.1 was employed to display protein-protein interactions (*Shannon et al., 2003*).

## IBAQ analysis

The IBAQ method was adapted from the original description (*Schwanhäusser et al., 2011*) for two independent whole cell analyses of wild-type (WT) HCMV strain Merlin infection at 24, 48 and 72 hr PI. These included: (a) WCL3 from *Weekes et al. (2014)* (conditions examined were 0, 24, 48, 72, 96 hr PI with WT Merlin with or without the viral DNA synthesis inhibitor phosphonoformate); (b) proteomic series three from *Fielding et al. (2017)* (0, 24, 48, 72 hr PI with WT Merlin with or without the lysosomal inhibitor leupeptin, or with an HCMV recombinant having a block deletion in the US12-US21 region). The maximum MS1 precursor intensity for each quantified peptide was determined for each experiment, and a summed MS1 precursor intensity for each protein across all matching peptides was calculated. To determine the proportion of the summed intensity that arose at 24, 48 and 72 hr PI, the summed intensity was adjusted in proportion to normalised TMT values: (24 hr + 48 hr + 72 hr PI) / $\sum$(all quantified times or conditions). Adjusted intensities were divided by the number of theoretical tryptic peptides from each protein between 7 and 30 amino acid residues in length to give an estimated IBAQ value. The same calculation was used to estimate IBAQ abundances for viral proteins (*Supplementary file 1A*, columns C-E) and human proteins (*Supplementary file 1C* columns E-G). Viral and human IBAQ values in these columns can be directly compared to examine the relative abundances of HCMV and host proteins.

Where PSMs had been assigned to a non-canonical viral ORF but were redundant to a canonical viral protein, peptides corresponding to the canonical protein were separated from those unique to the ORF, generating two separate entries as described in 'Interactor Identification with CompPASS'. For the non-canonical ORF, the number of theoretical peptides from the non-canonical protein fragment were used in the IBAQ calculation.

Values were separately normalised for HCMV and human proteins by the sum of all IBAQ values within each experiment, and average and range of the normalised values calculated and plotted (*Figure 1—figure supplement 1A*, *Supplementary file 1A* columns F-I, *Supplementary file 1C* columns H-K).

## Interaction database comparisons

For purposes of comparison, lists of physical interactions between viral proteins and human proteins were downloaded in October 2018 from: BioGRID (*Chatr-Aryamontri et al., 2013*), IntAct (*Orchard et al., 2014*), Uniprot (www.uniprot.org), MINT (*Licata et al., 2012*), and Virus Mentha (*Calderone et al., 2015*).

## Domain association analysis

Domain enrichments were calculated by mapping Pfam domains drawn from Uniprot onto human and HCMV amino acid sequences. The total number of interactions that included each domain, and the number of interactions involving pairs of domains whose parent proteins associate, were counted. The significance of the association among co-occurring domains was calculated using Fisher's Exact Test as described previously (*Huttlin et al., 2015*). p-values were corrected for multiple hypothesis testing (*Benjamini and Hochberg, 1995*). Domains were considered significantly associated if their adjusted p-value was <0.01. Overall 96 domains have been identified in HCMV proteins by Pfam, but only 10 domains were identified in two or more baits. Only this subset was examined in *Figure 4A* and *Supplementary file 5* to increase confidence in domain association predictions.

## Statistical analysis

*Figure 2*, *Figure 2—figure supplement 1* Benjamini-Hochberg adjusted p-values for enrichment are shown as blue surrounds to each pathway where p<0.05. More significantly enriched pathways are shown in darker blue as detailed in the figures.

*Figure 4* The significance of the association among co-occurring domains was calculated using Fisher's Exact Test as described previously (*Huttlin et al., 2015*). p-values were corrected for multiple hypothesis testing.

*Figure 5 (B)* Benjamini-Hochberg adjusted Significance A values were used to estimate p-values in the top panels; \*\*p<0.005, \*\*\*p<0.0005. Mean and SEM are shown for transcript quantitation (n = 3) in the middle panels. A p-value for the difference between rates of degradation is shown in the bottom panel; \*\*\*p<0.0005. All calculations and statistics are described in *Nightingale et al. (2018)*. (F) Mean and SEM are shown for transcript quantitation as in (B).

*Figure 6 (F)* p-values for a difference between wild-type and ORFL147C-deficient virus were estimated using a two-tailed Student's t-test. \*\*\*p<0.001, \*\*\*\*p<0.0001.

*Figure 7* Benjamini-Hochberg adjusted p-values are shown for each enriched pathway.

*Figure 1—figure supplement 1* Average IBAQ values + /- range are plotted for proteins quantified in both analyses (n = 2).

## Pathway analysis

The Database for Annotation, Visualisation and Integrated Discovery (DAVID) version 6.8 was used to determine pathway enrichment for *Figure 2* and *Figure 2—figure supplements 1* and *2A* (*Huang et al., 2009*), in which all human HCIPs for all viral baits were searched against a background of all human proteins, using default settings. For *Figure 6C* and *Figure 4—figure supplement 1*, DAVID and Reactome software (*Fabregat et al., 2018*) were used to analyse 80 human HCIPs interacting with ORFL147C compared to all human proteins as background.

To identify type I interferon-stimulated genes (ISG) for *Figure 3D*, gene symbols were searched in 'Interferome 2.0' (http://interferome.org/interferome/home.jspx) (*Rusinova et al., 2013*). A gene

was considered to be an ISG if it was upregulated at least 2-fold by type I interferon in at least two independent experiments in human cells.

## Data availability

The mass spectrometry proteomics data have been deposited to the ProteomeXchange Consortium (http://www.proteomexchange.org/) via the PRIDE (*Vizcaíno et al., 2016*) partner repository with the dataset identifier PXD014845.

## Acknowledgements

We are grateful to Prof. Steve Gygi for providing access to the 'MassPike' software pipeline for quantitative proteomics. This work was supported by a Wellcome Trust Senior Clinical Research Fellowship (108070/Z/15/Z) to MPW; MRC Project Grants to GWGW (MR/L018373/1, MR/P001602/1), RJS (MR/L018373/1, MR/S00971X/1) and ECWY (MR/L018373/1, MR/P001602/1, MR/S00971X/1); a Wellcome Trust Programme Grant (WT090323MA) to GWGW and an MRC Programme Grant (MC_UU_12014/3) to AJD. ELH was funded by NIH grant U24 HG006673. This study was additionally supported by the Cambridge Biomedical Research Centre, UK.

## Additional information

### Funding

| Funder | Grant reference number | Author |
|---|---|---|
| Wellcome | 108070/Z/15/Z | Michael P Weekes |
| Medical Research Council | MR/L018373/1 | Eddie CY Wang<br>Gavin WG Wilkinson<br>Richard J Stanton |
| Medical Research Council | MR/P001602/1 | Eddie CY Wang<br>Gavin WG Wilkinson |
| Wellcome | WT090323MA | Eddie CY Wang<br>Gavin WG Wilkinson<br>Richard J Stanton |
| Medical Research Council | MC_UU_12014/3 | Andrew J Davison |
| National Institutes of Health | U24 HG006673 | Edward L Huttlin |
| Medical Research Council | MR/S00971X/1 | Eddie CY Wang<br>Richard J Stanton |

The funders had no role in study design, data collection and interpretation, or the decision to submit the work for publication.

### Author contributions

Luis V Nobre, Resources, Data curation, Formal analysis, Validation, Investigation, Visualization, Methodology, Writing - review and editing; Katie Nightingale, Benjamin J Ravenhill, Data curation, Formal analysis, Validation, Investigation, Visualization, Final approval of the version to be published; Robin Antrobus, Formal analysis, Investigation; Lior Soday, Resources, Data curation, Software, Final approval of the version to be published; Jenna Nichols, Data curation, Investigation; James A Davies, Sepehr Seirafian, Resources, Final approval of the version to be published; Eddie CY Wang, Formal analysis, Writing - review and editing, Final approval of the version to be published; Andrew J Davison, Data curation, Writing - review and editing, Final approval of the version to be published; Gavin WG Wilkinson, Supervision, Funding acquisition, Investigation, Writing - review and editing; Richard J Stanton, Resources, Formal analysis, Supervision, Funding acquisition, Writing - review and editing, Final approval of the version to be published; Edward L Huttlin, Resources, Data curation, Software, Visualization, Methodology, Writing - review and editing, Final approval of the version to be published; Michael P Weekes, Conceptualization, Resources, Data curation, Formal analysis, Supervision,

Funding acquisition, Investigation, Visualization, Methodology, Writing - original draft, Project administration, Writing - review and editing, Final approval of the version to be published

### Author ORCIDs
Luis V Nobre ⬤ https://orcid.org/0000-0003-0467-8989
Katie Nightingale ⬤ https://orcid.org/0000-0001-9958-4699
Lior Soday ⬤ https://orcid.org/0000-0001-6927-2985
James A Davies ⬤ http://orcid.org/0000-0003-3569-4500
Eddie CY Wang ⬤ https://orcid.org/0000-0002-2243-4964
Andrew J Davison ⬤ https://orcid.org/0000-0002-4991-9128
Gavin WG Wilkinson ⬤ http://orcid.org/0000-0002-5623-0126
Richard J Stanton ⬤ http://orcid.org/0000-0002-6799-1182
Edward L Huttlin ⬤ https://orcid.org/0000-0002-1822-1173
Michael P Weekes ⬤ https://orcid.org/0000-0003-3196-5545

### Decision letter and Author response
Decision letter https://doi.org/10.7554/eLife.49894.sa1
Author response https://doi.org/10.7554/eLife.49894.sa2

## Additional files

### Supplementary files
• Supplementary file 1. Details of the interactome. (**A**) Relative abundance of all canonical and non-canonical viral proteins quantified in experiment whole cell lysate 3 (WCL3) from *Weekes et al. (2014)* and whole cell lysate series three from *Fielding et al. (2017)*. Further details of the calculations employed are given in *Figure 1—figure supplement 1A* and the Materials and methods section. (**B**) Details of all 172 baits. Bait expression was verified by IB, MS or RT-qPCR (*Figure 1—figure supplement 1B*). (**C**) Relative abundance of all human proteins expressed in HFFFs, calculated as described in (**A**). The 'rank' column indicates the ranked average IBAQ abundance. The most abundant protein calculated by this method was ranked 1, and least abundant ranked 8129. (**D**) Coding sequences of all viral genes used in this study. A six base-pair linker region, a V5 tag then a stop codon directly followed each sequence (Key Resources Table). Codon usage was optimised for expression for US14, US17 and UL74. (**E**) Oligonucleotides and templates employed in the generation and RT-qPCR of each viral vector. (**F**) Oligonucleotides and templates employed in the generation and RT-qPCR of each human overexpression vector.

• Supplementary file 2. Full interactome data. (**A**) Numbers of HCIPs per bait, excluding bait-bait interactions. (**B**) HCIPs for each bait (see *Figure 1* and the Materials and methods section for details of the filtering employed, and the scores shown in this table). For baits solubilised in NP40, VHCIPs are shown in green. The 'Prey IBAQ rank' column shows the ranked IBAQ abundance from *Supplementary file 1C*, and gives an indication of how abundant each prey protein was in infected HFFFs. A range of ranks is shown where more than one isoform of a protein could be detected, in order to reflect data for all isoforms of that protein. Abundantly expressed prey may be more easily validated using IB with antibodies against an endogenous protein; less abundant proteins may require overexpression to enable detection. (**C**) All detected interacting proteins for each bait, without filtering.

• Supplementary file 3. Validation of the interactome data from BioGRID, IntAct, Uniprot, MINT and Virus Mentha (*Calderone et al., 2015*; *Chatr-Aryamontri et al., 2013*; *Licata et al., 2012*; *Orchard et al., 2014*). Columns give details of the database(s) that included each interaction, the method used, and cell type employed. Interactome scores from the present study are shown in columns H-K. Column L shows whether a given interaction was validated in this interactome. A value of 1 indicates validation; 0 indicates detection of the interaction but failure to pass stringent scoring thresholds; 'ND' indicates the interaction was not detected by the interactome. Column M shows the ranked abundance of each human prey protein from *Supplementary file 1C*. Interactions that

were not detected in this study included a number of prey proteins that could not be detected in HFFFs. Further details are given in the Materials and methods section.

• Supplementary file 4. Enriched functional pathways, protein components and interacting viral baits. (**A**) All enriched functional pathways amongst all human HCIPs (p<0.05, after Benjamini-Hochberg adjustment). Column D shows the bait(s) interacting with each pathway component. (**B**) Further details of viral baits interacting with components of each pathway. Two values are shown: '% interaction', the percentage of human interactors of each bait that belonged to the pathway (relates to *Figure 2*, where viral baits are included if >33% of interactors belonged to a given pathway). '% function' illustrates the percentage of proteins from the pathway that interacted with the bait (relates to *Figure 2—figure supplement 1*, where viral baits are included if >33% of the pathway components identified interacted with a given viral bait). Values of >33% are coloured in this table. The 'count' column shows the total number of interacting pathway members; *Figure 2* and *Figure 2—figure supplement 1* included data with counts ≥ 2. (**C**) All enriched functional pathways amongst human HCIPs from each temporal class (p<0.05, after Benjamini-Hochberg adjustment). Column E shows the bait(s) interacting with each pathway component. This data underlies *Figure 2—figure supplement 2A*. (**D**) Temporal interactions of viral bait and viral prey proteins. This data underlies *Figure 2—figure supplement 2B*.

• Supplementary file 5. Full data underlying the domain-domain association predictions. (**A**) HCMV proteins that contain each described Pfam domain. Links are given to additional information on each domain on the Pfam website. Overall 96 domains have been identified in HCMV proteins by Pfam, however only 10 domains were identified in two or more baits. Only this subset was examined to increase confidence in domain association predictions. (**B**) Subset of *Supplementary file 2B* illustrating individual protein-protein interactions that underpin data shown in *Figure 4A*.

• Supplementary file 6. Proteins degraded early during HCMV infection from *Nightingale et al. (2018)*, using sensitive criteria. Interactome data identified viral baits for 31 of these degraded proteins.

• Supplementary file 7. Enrichment of functional pathways among proteins interacting with (**A**) ORFL147C, using DAVID software and a maximum p-value of 0.3; (**B**) ORFL147C, using the Reactome database and ≥4 entities per enriched pathway; (**C**) both HCMV and KSHV (*Davis et al., 2015*), using DAVID software and a maximum p-value of 0.05; (**D**) only HCMV as described in *Figure 5C*, using DAVID software and a maximum p-value of 0.01.

• Transparent reporting form

## Data availability

All data analysed during this study are included in the manuscript and supporting files. The mass spectrometry proteomics data have been deposited to the ProteomeXchange Consortium (http://www.proteomexchange.org/) via the PRIDE (Vizcaino et al., 2016) partner repository with the dataset identifier PXD014845.

The following dataset was generated:

| Author(s) | Year | Dataset title | Dataset URL | Database and Identifier |
|---|---|---|---|---|
| Nobre L, Nightingale K, Ravenhill B, Antrobus R, Soday L, Nichols J, Davies J, Wang ECY, Davison AJ, Wilkinson GWG, Stanton RJ, Huttlin EL, Weekes MP | 2019 | Global analysis of the human cytomegalovirus interactome identifies degradation hubs, domain associations and viral protein functions | http://proteomecentral.proteomexchange.org/cgi/GetDataset?ID=PXD014845 | ProteomeXchange, PXD014845 |

The following previously published dataset was used:

| Author(s) | Year | Dataset title | Dataset URL | Database and Identifier |
|---|---|---|---|---|
| Nightingale K, Lin | 2018 | High definition analysis of protein | http://proteomecentral. | ProteomeXchange, |

| KM, Ravenhill B, Ruckova E, Davies C, Nobre L, Fielding CA, Fletcher-Etherington A, Soday L, Nichols H, Sugrue D, Wang ECY, Moreno P, Umrania Y, Antrobus R, Davison AJ, Wilkinson GWG, Stanton RJ, Tomasec P, Weekes MP | stability during cytomegalovirus infection informs on cellular restriction | proteomexchange.org/cgi/GetDataset?ID=PXD009945 | PXD009945 |

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
