## [Decision Letter]

**Acceptance summary:**

The manuscript presents an impressive and elaborate amount of data and results about the human cytomegalovirus 'interactome' using immunoprecipitation-based mass spectrometry, revealing an extensive network of over 3400 high confidence interacting proteins and over 150 virus-virus protein interactions after IP mass spec experiments with 170 stable human cell lines derived from immortalized primary human fetal foreskin fibroblasts, each expressing a single V5 tagged-HCMV protein. The methods, data and results presented in the manuscript, are of high quality and of high value to people in the virology field and biologists overall. All comments raised by the reviewers have been answered in detail. This work provides a tremendous and comprehensive resource for future studies of this and related pathogens.

**Decision letter after peer review:**

Thank you for submitting your article "Human cytomegalovirus interactome analysis identifies degradation hubs, domain associations and viral protein functions" for consideration by *eLife*. Your article has been reviewed by three peer reviewers, including Piet Maes as the Reviewing Editor, and the evaluation has been overseen by Wendy Garrett as the Senior Editor. The following individual involved in review of your submission has agreed to reveal their identity: Moriah L Szpara.

The reviewers have discussed the reviews with one another and the Reviewing Editor has drafted this decision to help you prepare a revised submission.

Summary:

Luis Nobre and colleagues used a combination of stable cell lines expressing tagged viral proteins, native infection by human cytomegalovirus (HCMV), and immunoprecipitation (IP)-based mass spectrometry (MS) to collect data on human and viral proteins interacting with ~160 HCMV proteins. Peptides detected by IP-MS were subjected to extensive bioinformatics analyses thereafter, to detect high-confidence interactions, and to provide insights into the pathways and cellular programs impacted by those interactions. A small number of "wet bench" follow-up analyses provided confidence in the IP-MS interactions detected, and demonstrates an approach for further studies of specific interactions and pathways. The data quality is excellent, as is the subsequent analysis provided by the authors. The work provides a tremendous and comprehensive resource for future studies of this and related pathogens.

The reviewers raise a number of concerns that must be adequately addressed before the paper can be accepted. Some of the required revisions will likely require further experimentation within the framework of the presented study and used techniques.

Essential revisions:

1) The reviewers raised concerns about the approach described wherein the authors use pooled biological replicates for mass spec. Is averaging out biological variability a good approach? Pooling and then splitting would essentially average out the data. The only real significance in doing that approach would be to assess the instrument's reliability, which is not a major problem with mass spec, especially not with the newer instruments. Treating the biological replicates separately, running each one independently on the mass spec. would be better. This way, the reproducibility of the experiments for each bait is truly measured. If possible, we would recommend to rerun a small subset of the IP-MS/MS experiments, analysing each replicate independently, to demonstrate the robustness, reproducibility and reliability of the experiments.

2) Is the SFFV promoter used constitutive, and/or driving higher than usual expression? If so, can the authors comment on the potential implications of this for their data? The Discussion includes mention of the potential for missed interactions due to low levels of prey proteins. However the over-abundance, and/or temporal dysregulation of viral protein expression, might also be enabling interactions that wouldn't normally take place at that time point in infection. To what degree is that an issue? While this issue is not large enough to undercut the high value of the manuscript, it likely belongs in the Discussion.

3) The authors explain that HCMV has 5 distinct temporal gene expression classes, spanning over 96 hours post-infection, while the interactome described was determine at 60 hours post-infection. The authors make it clear why 60 hours post-infection was selected, but I would like to hear more from the authors about how the timing of viral protein expression may influence the interactome so that the data could be put into a more natural biological context. It would be very useful and potentially insightful to show what interactions viral proteins have as a function of time. For example, (a) are there major differences in the human protein networks interacting with viral proteins from each Tp stage, (b) does the nature of the human protein networks found to interact with proteins from each Tp stage help explain the viral life cycle, (c) how are the virus-virus protein interactions found to evolve as a function of the Tp stage – are they any of these interactions that do not make temporal sense?

4) I understand why the authors have attempted to validate protein-protein interactions identified by the IP-MS experiments, but I have concerns about the manner in which the experiments were performed. In my opinion, Co-IP Western blots to validate selected interactions should be done either using non-tagged proteins (if possible) or done using either a different epitope tag or placing the original tag at the opposite end of the bait protein. Otherwise, using the same tagged bait construct to "validate" what was already seen in the IP-MS experiments appears redundant, unnecessary, and does not address if the original interaction observed could have been due to an artefact of the system. The mass spec is perhaps the best way to determine if a protein is present and is often times more quantitative and sensitive than a Western blot, so essentially re-doing to IP-MS using an overexpressed, tagged prey protein, in my opinion, doesn't offer much validation. I understand that it may not be possible to re-do all of the validation experiments, but I would like to see, at least for the most important interactions the authors seek to validate, that these experiments be repeated using either a different tag on the bait or by using the V5 tag on the N-terminus instead.

5) With regards to the UL42 protein interactions with various E3's and the experiments shown to demonstrate that UL42 expression results in a decrease in NEDD4 protein levels, I have a couple of questions/comments. First, the authors show data that NEDD4 levels are decreased to a new steady state by 24 hours post-infection. Could the authors please comment on when UL42 is normally expressed after HCMV infection to clarify if normal UL42 expression correlates with the decrease in NEDD4 levels? My question is the same regarding NEDD4L as well. Second, the authors offer no explanation for how UL42 alters NEDD4 and NEDD4L levels – is this through the UPS, the lysosome, etc.? Adding some simple additional experiments could help shed some light on this (for example, does blocking the proteasome rescue NEDD4/4L loss?).

6) In the ORFL147C experiments, the authors suggest that the impairment to viral replication when ORFL147C is mutated could be due to its interaction with host cell splicing or transcriptional mechanisms. From the data shown, it appears that loss of ORFL147C (I would have liked to have a seen a WB, if possible, to show that ORFL147C is in fact no longer expressed) does decrease viral replication, but it is not at all clear if that has anything to do with the suggested host cell splicing or transcriptional mechanisms. In fact, given the large HCIP network for ORFL147C, there are a number of potential reasons for this phenotype. I agree that this protein appears to significantly effect viral replication, but I feel that the authors should specifically say that the mechanism of how ORFL147C plays a role in HMCV replication remains unknown.

---

## [Author Response]

Essential revisions:1) The reviewers raised concerns about the approach described wherein the authors use pooled biological replicates for mass spec. Is averaging out biological variability a good approach? Pooling and then splitting would essentially average out the data. The only real significance in doing that approach would be to assess the instrument's reliability, which is not a major problem with mass spec, especially not with the newer instruments. Treating the biological replicates separately, running each one independently on the mass spec. would be better. This way, the reproducibility of the experiments for each bait is truly measured. If possible, we would recommend to rerun a small subset of the IP-MS/MS experiments, analysing each replicate independently, to demonstrate the robustness, reproducibility and reliability of the experiments.

The aim of pooling the biological replicates was not to average out biological variability. Rather, there were two distinct rationales:

i) To solve specific technical issues of (a) the potential for carry-over of peptides between adjacent injections of different IP samples to the mass spectrometer and (b) to use consistency of detection of prey as a measure of confidence in bait-prey interaction (which is encapsulated in the ‘entropy’ score). Both of these issues are well understood and examined in Huttlin et al., 2015; the CompPass algorithm was developed with this specific protocol in mind. In our manuscript, carry-over was minimised by interspersing two ‘wash’ injections between adjacent samples, however particularly abundant peptide species in a given sample nevertheless had the potential to contaminate subsequent injections. The run order was therefore reversed from one batch of replicate analyses to the next to ensure that any carry-over was different in each case (i.e. Run 1: Sample A, wash, wash, Sample B, wash, wash, Sample C…; Run 2: …Sample C, wash, wash, Sample B, wash, wash, Sample A). To electronically filter out carry-over contaminants, the entropy score compared the number of peptide-spectrum matches (PSM) between technical replicate injections and eliminated prey that were not detected consistently. In addition, this form of replicate analysis also enabled false positive interactions to be minimised since the same random incorrect ID was unlikely to appear twice in two different runs. It was therefore important that replicate injection material was as similar as possible to ensure this filter was efficacious.

ii) To solve biochemical issues of sufficiency of the amount of injected material to ensure that enough was present for MS analysis after all purification steps had been completed. If true biological replicates had been used, technical replicates of each biological replicate would also have been required for the entropy score, which would have doubled the required MS instrument time (already 472 hours of injection plus 531 hours of washes) and doubled the required amount of virus.

We have added detail above to both the Materials and methods section and Figure 1—figure supplement 1D legend. To directly examine biological variability, we have re-run six of the IP-MS experiments as suggested, with independent analysis of each replicate. We found that there was very good correlation between the number of PSM from each identified prey protein between biological replicates, and have added this data to Figure 1—figure supplement 1D.

2) Is the SFFV promoter used constitutive, and/or driving higher than usual expression? If so, can the authors comment on the potential implications of this for their data? The Discussion includes mention of the potential for missed interactions due to low levels of prey proteins. However the over-abundance, and/or temporal dysregulation of viral protein expression, might also be enabling interactions that wouldn't normally take place at that time point in infection. To what degree is that an issue? While this issue is not large enough to undercut the high value of the manuscript, it likely belongs in the Discussion.

The SFFV promoter used did facilitate constitutive expression of the viral transgene, and we have modified the text in the Materials and methods section to make this explicitly clear. We agree that a potential caveat of the approach we employed is enabling interactions that might not usually take place at 60 h of infection. However, we attempted to minimise this possibility by selecting a time-point of infection at which we have good data suggesting every viral protein is expressed at some level, from our previous proteomic studies (Figure 1—figure supplement 1E). Temporal dysregulation of viral protein expression secondary to stable bait expression throughout the course of infection might also lead to observation of interactions at 60 h of infection that usually occurred at a different infection time point. These interactions would nevertheless be anticipated to take place at some point during viral infection, as their ‘survival’ as high-confidence interacting proteins after filtering is strongly suggestive of a genuine interaction.

Stable overexpression of the viral bait might similarly enable false interactions. However, for viral interactomes, certain proteins that are endogenously expressed by the virus (as opposed to the host) are already under the control of strong promoters (for example, Mocarski et al., 2013). Indeed, some viral bait proteins expressed using the lentiviral system may actually be expressed at a lower level in comparison to during HCMV infection. From our IBAQ analysis of host and viral protein abundance averaged across 24, 48 and 72 h of HCMV infection, the most abundant viral protein (UL83) was expressed at an ~2.4-fold greater level than the most abundant host protein (Galectin-1), and the least abundant viral protein at a ~62-fold greater level than the least abundant host protein (Supplementary file 1A and 1C) suggesting that the range of expression of viral proteins was already shifted towards the higher end of host protein expression.

Previous interactome studies have found no correlation between bait protein expression and the number of HCIP. For example, Sowa et al., 2009 found “Flag-HA-Dub expression varied over a wide range, yet no correlation was observed between Dub protein levels and their number of HCIPs …. (a correlation would be expected if overexpression consistently led to increases in non-physiological interactions)”.

Alternative strategies to conduct an interactome study would also suffer from potential confounding issues, albeit different ones. For example, introduction of a tag into the viral genome prior to or after each coding sequence may facilitate expression of the bait at the same time and level as during infection with unmodified virus. However, due to the frequent occurrence of polycistronic transcripts and overlapping ORFs (Stern-Ginossar, 2012), introduction of a tag may disrupt expression of neighbouring genes.

We have modified the Discussion to reflect these points:

“Overexpression of each bait throughout the course of infection may have led to temporal dysregulation of the expression of other viral proteins, and may have facilitated interactions that would usually commence earlier or later than 60 h of infection. […] However, due to the occurrence of polycistronic transcription of viral genes and overlapping viral ORFs (Stern-Ginossar et al., 2012), introduction of a tag may disrupt expression of neighbouring genes.”

3) The authors explain that HCMV has 5 distinct temporal gene expression classes, spanning over 96 hours post-infection, while the interactome described was determine at 60 hours post-infection. The authors make it clear why 60 hours post-infection was selected, but I would like to hear more from the authors about how the timing of viral protein expression may influence the interactome so that the data could be put into a more natural biological context. It would be very useful and potentially insightful to show what interactions viral proteins have as a function of time. For example, (a) are there major differences in the human protein networks interacting with viral proteins from each Tp stage, (b) does the nature of the human protein networks found to interact with proteins from each Tp stage help explain the viral life cycle, (c) how are the virus-virus protein interactions found to evolve as a function of the Tp stage – are they any of these interactions that do not make temporal sense?

Very many thanks for this interesting suggestion. We have now conducted DAVID enrichment analysis of human HCIPs for each of the five Tp classes of viral protein expression, in comparison to all HCIPs for the whole interactome. Interestingly, there is a clear relation to functions required at different stages of the viral life-cycle. For example, Tp1 and Tp2 protein HCIPs are enriched in NuRD complex members, proteins involved in histone deacetylation and proteins with SANT domains (which function in chromatin remodelling). Tp3 HCIPs are enriched in functions required for viral genomic replication and immune evasion, whilst Tp5 HCIPs are enriched in functions directed at intracellular trafficking and secretion. Furthermore, we have analysed viral HCIPs to determine whether viral-viral protein-protein interactions predominantly derive from the same Tp class. Two patterns emerged – (a) interaction of viral proteins within the same temporal class, or between adjacent classes; (b) interaction of proteins from the largest class (Tp5) with members of all of the different classes of viral protein. For example, Tp1 and Tp2 class proteins UL29 and UL38 interacted, as previously reported (Supplementary file 3, Figure 2). Tp1-class tegument proteins US23 and US24 interacted. The majority of Tp5 interactions were with other Tp5 proteins, 15/37 of which were tegument-tegument, capsid-capsid or tegument-capsid protein interactions (new Figure 2—figure supplement 2A, new Supplementary file 4C). Certain interactions between proteins in different temporal classes have also been reported; for example, between the Tp5 DNA polymerase accessory protein UL44 and Tp2 DNA polymerase UL54. Clearly, other novel interactions additionally exist between quite distinctly expressed proteins, for example between the functionally unknown Tp2-class membrane protein UL14 and two Tp5-class proteins: membrane protein UL121 and envelope glycoprotein UL4. We have added all of this data to a new supplementary figure, Figure 2—figure supplement 2A-B and Supplementary file 4C-D.

4) I understand why the authors have attempted to validate protein-protein interactions identified by the IP-MS experiments, but I have concerns about the manner in which the experiments were performed. In my opinion, Co-IP Western blots to validate selected interactions should be done either using non-tagged proteins (if possible) or done using either a different epitope tag or placing the original tag at the opposite end of the bait protein. Otherwise, using the same tagged bait construct to "validate" what was already seen in the IP-MS experiments appears redundant, unnecessary, and does not address if the original interaction observed could have been due to an artefact of the system. The mass spec is perhaps the best way to determine if a protein is present and is often times more quantitative and sensitive than a Western blot, so essentially re-doing to IP-MS using an overexpressed, tagged prey protein, in my opinion, doesn't offer much validation. I understand that it may not be possible to re-do all of the validation experiments, but I would like to see, at least for the most important interactions the authors seek to validate, that these experiments be repeated using either a different tag on the bait or by using the V5 tag on the N-terminus instead.

We agree with the reviewer’s point, however unfortunately very few antibodies against HCMV proteins themselves are commercially available for IP (only for UL83, UL55, UL123 and UL75). The V5 tag itself is only 14aa long and as such is unlikely to provide substantial steric hindrance to protein-protein interactions, although clearly has the potential to inhibit interactions involving the tagged bait C-terminus. To validate some of our most important interactions as requested, we have repeated a subset of co-IP immunoblots using N-terminally V5-tagged baits for UL42 and UL72, and have included the UL42 data in Figure 5—figure supplement 1.

Unfortunately, in the time available for revisions, N-terminally tagged UL72 itself could not be detected by immunoblot either in the input or IP samples. Nevertheless, CNOT2 was still detected in the UL72 IP (but not either control), suggesting that V5-UL72 was expressed albeit at low level (Author response image 1, representative of n=2 experiments). For this reason, we have not included the figure in the manuscript. Of note, we have already validated interaction of UL72-V5 both with CNOT2-HA, CNOT7-HA and both endogenously-expressed CNOT proteins, further increasing confidence in these observations (Figures 3A-B).

5) With regards to the UL42 protein interactions with various E3's and the experiments shown to demonstrate that UL42 expression results in a decrease in NEDD4 protein levels, I have a couple of questions/comments. First, the authors show data that NEDD4 levels are decreased to a new steady state by 24 hours post-infection. Could the authors please comment on when UL42 is normally expressed after HCMV infection to clarify if normal UL42 expression correlates with the decrease in NEDD4 levels? My question is the same regarding NEDD4L as well. Second, the authors offer no explanation for how UL42 alters NEDD4 and NEDD4L levels – is this through the UPS, the lysosome, etc.? Adding some simple additional experiments could help shed some light on this (for example, does blocking the proteasome rescue NEDD4/4L loss?).

We have not been able to detect UL42 protein in any of our published or unpublished proteomic experiments, likely at least partly due to the presence of only a single lysine, and only five arginine residues in the 125 amino acid protein. We did not quantify UL42 transcript in our previous RNAseq experiment during HCMV infection (Nightingale et al., 2018). No anti-UL42 antibody is commercially available. However, UL42 transcript was detected by Stern-Ginossar et al. in their 2012 Science manuscript (data added to Figure 5C). Although expression of this transcript peaked at 72 h of infection, it was nevertheless clearly detectable at 5 and 24 h post infection suggesting that UL42 protein is likely to be expressed contemporaneously with degradation of NEDD4 and NEDD4L.

Expression of NEDD4 and NEDD4L were rescued by application of both MG132 and Leupeptin during HCMV infection, as we show in Figure 5B. MG132 is known to affect lysosomal cathepsins in addition to the proteasome (Wiertz et al., 1996), and leupeptin is a naturally occurring protease inhibitor that can inhibit some proteasomal proteases in addition to the lysosome, making it difficult to definitively characterise the pathway of degradation from this data.

Unfortunately in the time available for revisions, we were unable to rescue expression of either NEDD4-HA or NEDD4L-HA with inhibitors of the proteasome or lysosome, in HFFF-TERTs stably expressing UL42 (Author response image 2, representative of n=4 experiments). This was despite optimisation both of timing of the analysis and inhibitor concentration.

**Author response image 2. respfig2:** 

This finding is consistent with results from Koshizuka et al., 2016, who characterised the degradation of ITCH by UL42 and found “In order to elucidate the stability of UL42, recombinant HCMV-infected cells were treated with chloroquine and MG132. HA-UL42 was increased in the presence of chloroquine, a lysosome inhibitor, but not in the presence of MG132, a proteasomal inhibitor, indicating that UL42 was degraded by the lysosome (Figure 6C). However, the band pattern of Itch was not significantly changed even in the presence of the inhibitors’. We have therefore added to the Results section “The route of degradation of each of the UL42 targets requires further characterisation. MG132 and leupeptin both inhibited degradation of each protein (Figure 5B), which may correspond to the known effects of MG132 on lysosomal cathepsins in addition to the proteasome (Wiertz et al., 1996), or effects of leupeptin on certain proteasomal proteases in addition to lysosomal proteases.”

6) In the ORFL147C experiments, the authors suggest that the impairment to viral replication when ORFL147C is mutated could be due to its interaction with host cell splicing or transcriptional mechanisms. From the data shown, it appears that loss of ORFL147C (I would have liked to have a seen a WB, if possible, to show that ORFL147C is in fact no longer expressed) does decrease viral replication, but it is not at all clear if that has anything to do with the suggested host cell splicing or transcriptional mechanisms. In fact, given the large HCIP network for ORFL147C, there are a number of potential reasons for this phenotype. I agree that this protein appears to significantly effect viral replication, but I feel that the authors should specifically say that the mechanism of how ORFL147C plays a role in HMCV replication remains unknown.

Many thanks for this suggestion. No ORFL147C antibody is available, however we are routinely able to detect this protein using proteomic approaches. We conducted a 3-plex tandem mass tag-based experiment comparing infection with mock, wild-type virus or the ORFL147C deletion mutant (48 h, MOI = 2). ORFL147C was expressed at the level of noise in mock- and ORFL147C-infected cells confirming deletion of this gene in the mutant. We have added this data to Figure 6G. We have added detail as suggested to say that the mechanism of how ORFL147C plays a role in HCMV replication remains unknown.